# Efficient Online Reinforcement Learning for Diffusion Policy

**Haitong Ma**[1]  **Tianyi Chen**[2]  **Kai Wang**[2]  **Na Li**[* 1]  **Bo Dai**[* 2]

## Abstract

Diffusion policies have achieved superior performance in imitation learning and offline reinforcement learning (RL) due to their rich expressiveness. However, the conventional diffusion training procedure requires samples from target distribution, which is impossible in online RL since we cannot sample from the optimal policy. Backpropagating policy gradient through the diffusion process incurs huge computational costs and instability, thus being expensive and not scalable. To enable efficient training of diffusion policies in online RL, we generalize the conventional denoising score matching by reweighting the loss function. The resulting Reweighted Score Matching (`RSM`) preserves the optimal solution and low computational cost of denoising score matching, while eliminating the need to sample from the target distribution and allowing learning to optimize value functions. We introduce two tractable reweighted loss functions to solve two commonly used policy optimization problems, policy mirror descent and max-entropy policy, resulting in two practical algorithms named Diffusion Policy Mirror Descent (`DPMD`) and Soft Diffusion Actor-Critic (`SDAC`). We conducted comprehensive comparisons on MuJoCo benchmarks. The empirical results show that the proposed algorithms outperform recent diffusion-policy online RLs on most tasks, and the `DPMD` improves more than 120% over soft actor-critic on Humanoid and Ant.

## 1. Introduction

Many successes of diffusion-based generative models have been witnessed recently (Sohl-Dickstein et al., 2015; Song & Ermon, 2019; Ho et al., 2020). With the iterative denoising design, diffusion models achieved superior expressiveness and multimodality in representing complex probability distributions, demonstrating remarkable performance in image and video generation (Ramesh et al., 2021; Saharia et al., 2022). The superior expressiveness and multimodality naturally benefit the policies in sequential decision-making problems. In fact, diffusion policy has been introduced in imitation learning and offline reinforcement learning (RL), where expert datasets are presented. Diffusion policies improved significantly over previous deterministic or unimodal policies on manipulation (Chi et al., 2023; Ke et al., 2024; Scheikl et al., 2024) and locomotion tasks (Huang et al., 2024).

Meanwhile, online RL has been long seeking expressive policy families. One promising direction lies in energy-based models (EBMs)—a class of probabilistic models that represent distributions via unnormalized densities. When applied to RL, energy-based policies—policies modeled as EBMs—have been shown to arise as optimal solutions in proximity-based policy optimization (Nachum et al., 2017; Mei et al., 2020) and max-entropy RL (Neu et al., 2017; Haarnoja et al., 2017). Despite their theoretical appeal, training and sampling from such unnormalized models in continuous action spaces are notoriously difficult due to their intractable likelihood (Song & Kingma, 2021). To mitigate this, a variety of probabilistic models have been introduced for efficient sampling and learning, but with the cost of approximation error. In practice, many algorithms (Schulman, 2015; Schulman et al., 2017; Haarnoja et al., 2018; Hansen-Estruch et al., 2023) project the energy-based policies onto the Gaussian policies. However, this projection severely limits the expressiveness of the original energy-based formulations, often resulting in degraded performance.

Diffusion models are closely related to EBMs, as they can be regarded as EBMs perturbed by a series of noise (Song & Ermon, 2019; Shribak et al., 2024), thus being the perfect candidate to represent energy-based policies in RL. Unfortunately, it is highly non-trivial to train diffusion policies in online RL. The commonly used diffusion model training procedure, denoising score matching (Ho et al., 2020), requires data samples from the target data distribution (usually a large image dataset in image generation). However,

---
[*]Equal supervision  [1]Harvard University.  [2]Georgia Institute of Technology.  Emails: Haitong Ma <haitongma@g.harvard.edu>, Na Li <nali@seas.harvard.edu>, Bo Dai <bodai@cc.gatech.edu>.

*Proceedings of the 42ⁿᵈ International Conference on Machine Learning*, Vancouver, Canada. PMLR 267, 2025. Copyright 2025 by the author(s).

we cannot sample from the optimal policy in online RL, where the policy is learned by optimizing the returns or value functions. There exist several preliminary studies trying to bypass the sampling issue (Psenka et al., 2023; Jain et al., 2024; Yang et al., 2023; Wang et al., 2024; Ding et al., 2024a; Ren et al., 2024), but all these methods suffer from biased estimations and/or huge memory and computation costs, resulting in suboptimal policies and limiting the true potential of diffusion policies in online RL.

To handle these challenges, we propose to generalize diffusion model training by reweighting the conventional denoising score matching loss, resulting in two efficient algorithms to train diffusion policies in online RL without sampling from optimal policies. Specifically,

• Building upon the viewpoint of diffusion models as noise-perturbed EBMs, we propose Reweighted Score Matching (RSM), a family of loss functions to train diffusion models, which generalizes the denoising score matching by reweighting the loss function while preserving the optimal solution as noise-perturbed EBMs.

• RSM leads to computationally tractable and efficient algorithms to train diffusion policies in online RL. We show that, by choosing different reweighting functions, we can train diffusion policies to solve two policy optimization problems, *policy mirror descent and max-entropy policy*, resulting in two practical algorithms named Diffusion Policy Mirror Descent (DPMD) and Soft Diffusion Actor-Critic (SDAC). Both problems are commonly seen in theoretical studies but empirically challenging in the continuous action space, and the proposed algorithms bridge this gap between the theory and practice of online RL.

• We conduct extensive empirical evaluation on MuJoCo, showing that the proposed algorithms outperform recent diffusion-based online RL baselines in most tasks. Moreover, both algorithms improve more than 100% over SAC on Humanoid and DPMD improves more than 100% over SAC on Ant, demonstrating the potential of diffusion policy in online RL.

## 2. Preliminaries

We introduce the necessary preliminaries in this section. First, we introduce reinforcement learning and two commonly seen policy optimization problems in online RL. Then we briefly recap the diffusion models and energy-based models.

### 2.1. Reinforcement Learning

**Markov Decision Processes (MDPs).** We consider Markov decision process (Puterman, 2014) specified by a tuple $\mathcal{M} = (\mathcal{S}, \mathcal{A}, r, P, \mu_0, \gamma)$, where $\mathcal{S}$ is the state space,

$\mathcal{A}$ is the action space, $r : \mathcal{S} \times \mathcal{A} \to \mathbb{R}$ is a reward function, $P : \mathcal{S} \times \mathcal{A} \to \Delta(\mathcal{S})$ is the transition operator with $\Delta(\mathcal{S})$ as the family of distributions over $\mathcal{S}, \mu_0 \in \Delta(\mathcal{S})$ is the initial distribution and $\gamma \in (0, 1)$ is the discount factor. We consider two types of commonly seen policy optimization problems in RL, (a) Policy mirror descent and (b) Max-entropy policy.

**Policy Mirror Descent** is closely related to practical proximity-based algorithms such as TRPO (Schulman, 2015) and PPO (Schulman et al., 2017), but with a different approach to enforce the proximity constraints. We consider policy mirror descent with Kullback–Leibler (KL) divergence proximal term (Tomar et al., 2021; Lan, 2023; Peters et al., 2010) updates the policy with

$$\pi_{\mathrm{MD}}(\boldsymbol{a}|\boldsymbol{s}) = \underset{\pi:\mathcal{S}\to\Delta(\mathcal{A})}{\operatorname{argmax}} \ \mathbb{E}_{\boldsymbol{a}\sim\pi}\left[Q^{\pi_{\mathrm{old}}}(\boldsymbol{s},\boldsymbol{a})\right] - \lambda D_{KL}\left(\pi||\pi_{\mathrm{old}};\boldsymbol{s}\right)$$

$$(1)$$

where $Q^{\pi_{\mathrm{old}}}(s,a) = \mathbb{E}_{\pi_{\mathrm{old}}}[\sum_{\tau=0}^{\infty} \gamma^{\tau} r(\boldsymbol{s}_t, \boldsymbol{a}_t)|\boldsymbol{s}_0 = s, \boldsymbol{a}_0 = a]$ is the state-action value function and $\pi_{\mathrm{old}}$ is the current policy. The additional KL divergence objective constrains the updated policy to be approximately within the trust region. The closed-form solution of policy mirror descent (1) satisfies

$$\pi_{\mathrm{MD}}\left(\boldsymbol{a}|\boldsymbol{s}\right) = \pi_{\mathrm{old}}\left(\boldsymbol{a}|\boldsymbol{s}\right) \frac{\exp\left(Q^{\pi_{\mathrm{old}}}\left(\boldsymbol{s},\boldsymbol{a}\right)/\lambda\right)}{Z_{\mathrm{MD}}(\boldsymbol{s})}, \quad (2)$$

and $Z_{\mathrm{MD}}(\boldsymbol{s}) = \int \pi_{\mathrm{old}}\left(\boldsymbol{a}|\boldsymbol{s}\right)\exp\left(Q\left(\boldsymbol{s},\boldsymbol{a}\right)/\lambda\right)d\boldsymbol{a}$ is the partition function.

**Max-entropy RL.** Maximum entropy RL considers the entropy-regularized expected return as the policy learning objective to justify the optimal stochastic policy

$$\arg\max_{\pi} J(\pi) := \mathbb{E}_{\pi}\left[\sum_{\tau=0}^{\infty} \gamma^{\tau}\left(r(\boldsymbol{s}_{\tau}, \boldsymbol{a}_{\tau}) + \lambda\mathcal{H}(\pi(\cdot|\boldsymbol{s}_{\tau}))\right)\right]$$

$$(3)$$

where $\mathcal{H}\left(\pi(\cdot|\boldsymbol{s})\right) = \mathbb{E}_{\boldsymbol{a}\sim\pi(\cdot|\boldsymbol{s})}[-\log\pi(\boldsymbol{a}|\boldsymbol{s})]$ is the entropy, $\lambda$ is a regularization coefficient for the entropy. The soft policy iteration algorithm (Haarnoja et al., 2017) is proposed to solve the optimal max-entropy policy. Soft policy iteration algorithm iteratively conducts soft policy evaluation and soft policy improvement, where soft policy evaluation updates the soft $Q$-function by repeatedly applying soft Bellman update operator $\mathcal{T}^{\pi}$ to current value function $Q : \mathcal{S} \times \mathcal{A} \to \mathbb{R}$, *i.e.*,

$$\mathcal{T}^{\pi}Q(\boldsymbol{s}_{\tau}, \boldsymbol{a}_{\tau}) = r(\boldsymbol{s}_{\tau}, \boldsymbol{a}_{\tau}) + \gamma\mathbb{E}_{\boldsymbol{s}_{\tau+1}\sim P}\left[V(\boldsymbol{s}_{\tau+1})\right] \quad (4)$$

where $V(\boldsymbol{s}_{\tau}) = \mathbb{E}_{\boldsymbol{a}_{\tau}\sim\pi}\left[Q(\boldsymbol{s}_{\tau}, \boldsymbol{a}_{\tau}) - \lambda\log\pi(\boldsymbol{a}_{\tau}\mid\boldsymbol{s}_{\tau})\right]$. Then in the soft policy improvement stage, the policy is updated to fit the target max-entropy policy

$$\pi_{\mathrm{MaxEnt}}(\boldsymbol{a}|\boldsymbol{s}) = \frac{\exp\left(Q^{\pi_{\mathrm{old}}}\left(\boldsymbol{s},\boldsymbol{a}\right)/\lambda\right)}{Z(\boldsymbol{s})} \quad (5)$$

where $Q^{\pi_{\mathrm{old}}}\left(\boldsymbol{s},\boldsymbol{a}\right)$ is the converged result of (4) with $\mathcal{T}^{\pi_{\mathrm{old}}}$, $Z(\boldsymbol{s}) = \int \exp(Q^{\pi_{\mathrm{old}}}(\boldsymbol{s},\boldsymbol{a})/\lambda)d\boldsymbol{a}$. Max-entropy RL shows a foundational concept in the exploration-exploitation trade-

off with stochastic policies, leading to practical algorithms with strong performance even with the restrictive Gaussian policies such as soft actor-critic (Haarnoja et al., 2018).

## 2.2. Energy-Based Models

The closed-form solutions of both policy mirror descent (2) and max-entropy policy (5) have unknown normalization constants. Such probabilistic models with unknown normalization constants are known as energy-based models (EBMs), whose density functions can be abstracted as

$$p_0(\mathbf{x}) = \frac{\exp(-E(\mathbf{x}))}{Z}$$

where $Z = \int \exp(-E(\mathbf{x})) \, d\mathbf{x}$ is the unknown normalization constant or partition function. We only know the *energy functions* $E(\mathbf{x}) = -\log p_0(\mathbf{x})$, *i.e.*, the negative log density. The gradient of log density $\nabla_{\mathbf{x}} \log p_0(\mathbf{x}) = -\nabla_{\mathbf{x}} E(\mathbf{x})$ is called the *score functions*.

The unknown normalization constants $Z$ raise difficulties in training and sampling of EBMs (Song & Kingma, 2021). One of the commonly used approaches is the *score-based methods*, which first learns the score function via score matching (Hyvärinen & Dayan, 2005; Song et al., 2020) and then draws samples via Markov chain Monte Carlo (MCMC) such as Langevin dynamics with the learned score functions (Neal et al., 2011). However, the MCMC sampling is inefficient due to the lack of finite-time guarantees, preventing score-based EBMs from being widely used in practice.

In the practice of online RL, projection onto Gaussian policies is commonly used in policy optimization with EBMs. For example, the well-known soft actor-critic (SAC, Haarnoja et al., 2018) parameterize the policy as Guassian $\pi_\theta(a|s) = \mathcal{N}\left(\mu_{\theta_1}(s), \sigma^2_{\theta_2}(s)\right)$ and updates the parameters $\theta = [\theta_1, \theta_2]$ by optimizing the $KL$-divergence to the target max-entropy policy $\min_\theta D_{\mathrm{KL}}(\pi_\theta \| \pi_{\mathrm{MaxEnt}})$. However, the projection loses expressiveness, and the resulting policies might be sub-optimal, leaving a huge gap between theory and practice of energy-based policies.

## 2.3. Denoising Diffusion Probabilistic Models

Denoising diffusion probabilistic models (DDPMs, Sohl-Dickstein et al., 2015; Ho et al., 2020) are composed of a forward diffusion process that gradually perturbs the data distribution $\mathbf{x}_0 \sim p_0$ to a noise distribution $\mathbf{x}_T \sim p_T$, and a reverse diffusion process that reconstructs the data distribution $p_0$ from the noise distribution $p_T$. The forward corruption kernels are Gaussian with a variance schedule $\beta_1, \ldots, \beta_T$, resulting in the forward trajectories with joint distributions

$$q_{0:T}(\mathbf{x}_{0:T}) = p_0(\mathbf{x}_0) \prod_{t=1}^{T} q_{t|t-1}(\mathbf{x}_t|\mathbf{x}_{t-1})$$

where $q_{t|t-1}(\mathbf{x}_t|\mathbf{x}_{t-1}) := \mathcal{N}(\mathbf{x}_t; \sqrt{1-\beta_t}\mathbf{x}_{t-1}, \beta_t \mathbf{I})$, $\mathbf{x}_t$ is the perturbed data at $t$ step, and $p, q$ are probability distributions[1]. As the perturbations at every step are independent and additive Gaussian, we can directly sample the $t-$step perturbed data $\mathbf{x}_t$ by

$$\mathbf{x}_t \sim p_t(\mathbf{x}_t) = \int p_0(\mathbf{x}_0) q_{t|0}(\mathbf{x}_t|\mathbf{x}_0) d\mathbf{x}_0,$$

where $q_{t|0}(\mathbf{x}_t|\mathbf{x}_0) := \mathcal{N}(\mathbf{x}_t; \sqrt{\bar{\alpha}_t}\mathbf{x}_0 (1-\bar{\alpha}_t) \mathbf{I}), \bar{\alpha}_t = \prod_{l=1}^{t}(1-\beta_l)$
$$\tag{6}$$

The backward process recovers the data distribution from a noise distribution $p_T$ with a series of reverse kernels $p_{t-1|t}(\mathbf{x}_{t-1}|\mathbf{x}_t)$. The kernel of reverse process is also Gaussian and can be parametrized as $\mathcal{N}\left(\frac{1}{\sqrt{\alpha_t}}(\mathbf{x}_t + \beta_t s_\theta(\mathbf{x}_t; t)), \sigma_t^2 \mathbf{I}\right)$ with score networks $s_\theta(\mathbf{x}_t, t)$ and fixed covariance $\sigma_t = \frac{1-\bar{\alpha}_{t-1}}{1-\bar{\alpha}_t}\beta_t$ The score network $s_\theta(\mathbf{x}_t; t)$ is trained by the denoising score matching to match the forward and reverse processes (Ho et al., 2020), whose loss function is

$$\frac{1}{T}\sum_{t=0}^{T}(1-\bar{\alpha}_t) \underset{\substack{\mathbf{x}_0 \sim p_0 \\ \mathbf{x}_t \sim q_{t|0}}}{\mathbb{E}} \left[\left\| s_\theta(\mathbf{x}_t; t) - \nabla_{\mathbf{x}_t} \log q_{t|0}(\mathbf{x}_t|\mathbf{x}_0)\right\|^2\right]$$
$$\tag{7}$$

. The score function $\nabla_{\mathbf{x}_t} \log q_{t|0}(\mathbf{x}_t|\mathbf{x}_0)$ can be computed from the sampled Gaussian noise perturbing $a_0$ to be $a_t$, thus the loss in (7) is tractable and easy to implement. After learning the $s_\theta$ via (7), we can draw samples via the reverse diffusion process by the iterative formulation

$$\mathbf{x}_{t-1} = \frac{1}{\sqrt{\alpha_t}}(\mathbf{x}_t + \beta_t s_\theta(\mathbf{x}_t, t)) + \frac{1-\bar{\alpha}_{t-1}}{1-\bar{\alpha}_t}\beta_t \mathbf{z}_t \tag{8}$$

for $t = T, T-1, \ldots, 1$ and $\mathbf{z}_t \sim \mathcal{N}(0, \mathbf{I})$.

## 3. Reweighted Score Matching: A General Loss Family for Diffusion Models

In this section, we first present the connection between energy-based models and diffusion models, justifying the expressiveness of diffusion policy to represent energy-based policies such as (2),(5). Then we identify the difficulties in training of diffusion policy in the context of online RL, where the conventional denoising score matching is intractable. To mitigate this, we propose our core contribution, Reweighted Score Matching (RSM), by reweighting the denoising score matching loss.

**Notation.** To fit the diffusion policy context, we consider the diffusion policy notations, where the diffusion is on actions $a$ conditioned on state $s$. The score function has the additional input of states $s_\theta(a_t; s, t)$. The data distribu-

---

[1]We use $p$ and $q$ interchangeably as density function in this paper. Generally, $p$ represents intractable distributions (like the t-step marginal $p_t(\mathbf{x}_t)$), and $q$ represents tractable distributions such as the Gaussian corruption $q_{t|t-1}(\mathbf{x}_t|\mathbf{x}_{t-1})$.

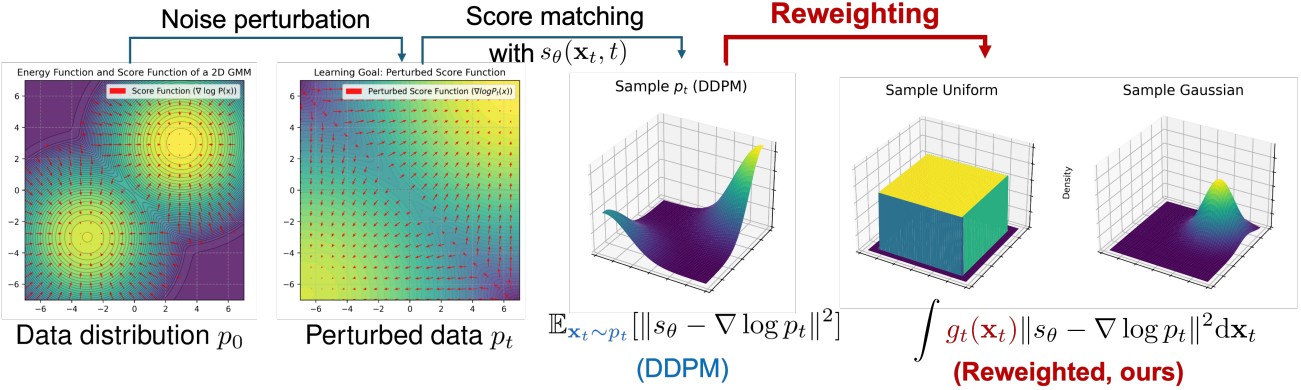

*Figure 1.* Diffusion model aims to match score network $s_\theta(\mathbf{x}_t, t)$ with noise-perturbed score function $\nabla_{\mathbf{x}_t} \log p_t(\mathbf{x}_t)$ by minimizing the expectation of error L2-norm $\|s_\theta(\mathbf{x}_t, t) - \nabla_{\mathbf{x}_t} \log p_t(\mathbf{x}_t)\|^2$ over distribution $p_t$. RSM generalize to other weight function $g_t$ to enable diffusion policy training in online RL.

tion $p_0(\cdot|\boldsymbol{s})$ refers to the policy in (2) or (5), and we will explicitly mention which one we refer to as $p_0$ if needed.

### 3.1. Diffusion Models as Noise-Perturbed Energy-Based Models

We first revisit the energy-based view of diffusion models, *i.e.*, *diffusion models are noise-perturbed EBMs* (Song & Ermon, 2019; Shribak et al., 2024), to justify that the diffusion policy can efficiently represent the energy-based policies.

**Proposition 3.1** (Diffusion models as noise-perturbed EBMs)**.** *Consider a single term in loss function of DDPM (7) at given state $\boldsymbol{s}$ and time $t$,*

$$\mathcal{L}_{\mathrm{DSM}}(\theta; \boldsymbol{s}, t) :=$$
$$\mathop{\mathbb{E}}_{\substack{\boldsymbol{a}_0 \sim p_0(\cdot|s) \\ \boldsymbol{a}_t \sim q_{t|0}(\cdot|\boldsymbol{a}_0)}} \left[ \left\| s_\theta\left(\boldsymbol{a}_t; \boldsymbol{s}, t\right) - \nabla_{\mathbf{x}_t} \log q_{t|0}(\boldsymbol{a}_t|\boldsymbol{a}_0) \right\|^2 \right] \quad (9)$$

*We refer to it as the denoising score matching (DSM) loss. The optimal solution $\theta^*$ is achieved when the following holds for all $\boldsymbol{a}_t$*

$$s_{\theta^*}(\boldsymbol{a}_t; \boldsymbol{s}, t) = \nabla_{\boldsymbol{a}_t} \log p_t(\boldsymbol{a}_t|\boldsymbol{s})$$

*where $p_t(\boldsymbol{a}_t|\boldsymbol{s}) = \int q_{t|0}(\boldsymbol{a}_t|\boldsymbol{a}_0)p_0(\boldsymbol{a}_0|\boldsymbol{s})d\boldsymbol{a}_0$ is the noise-perturbed policy with perturbation kernel $q_{t|0}(\boldsymbol{a}_t|\boldsymbol{a}_0) = \mathcal{N}(\boldsymbol{a}_t; \sqrt{\bar{\alpha}_t}\boldsymbol{a}_0, (1 - \bar{\alpha}_t)\mathbf{I})$, for noise schedule index $t = 1, 2, \ldots, T$.*

The connection is first revealed in Vincent (2011), and we revisit it in this paper.

*Proof.* Consider the following loss function minimizing the squared error between $s_\theta(\boldsymbol{a}_t; \boldsymbol{s}, t)$ and $\nabla_{\boldsymbol{a}_t} \log p_t(\boldsymbol{a}_t|\boldsymbol{s})$,

$$\mathcal{L}_{\mathrm{VSM}}(\theta; \boldsymbol{s}, t) = \mathop{\mathbb{E}}_{\boldsymbol{a}_t \sim p_t(\cdot|\boldsymbol{s})} \left[ \|s_\theta(\boldsymbol{a}_t; \boldsymbol{s}, t) - \nabla_{\boldsymbol{a}_t} \log p_t(\boldsymbol{a}_t|\boldsymbol{s})\|^2 \right]$$
$$(10)$$

where we refer to it as vanilla score matching (VSM) loss. It is obvious that the minimizer of $\mathcal{L}_{\mathrm{VSM}}$ is $s_{\theta^*}(\boldsymbol{a}_t; \boldsymbol{s}, t) = \nabla_{\boldsymbol{a}_t} \log p_t(\boldsymbol{a}_t|\boldsymbol{s})$ for any $\boldsymbol{a}_t$. Then we show the following loss equivalence,

$$\mathcal{L}_{\mathrm{DSM}}(\theta; \boldsymbol{s}, t) = \mathcal{L}_{\mathrm{VSM}}(\theta; \boldsymbol{s}, t) + \texttt{constant} \quad (11)$$

where `constant` is a constant irrelevant with $\theta$. The detailed derivations are deferred to Appendix B.1.

Therefore, minimizing $\mathcal{L}_{\mathrm{DSM}}$ in DDPMs is equivalent to minimizing $\mathcal{L}_{\mathrm{VSM}}$, whose optimal solution is $s_{\theta^*}(\boldsymbol{a}_t; \boldsymbol{s}, t) = \nabla_{\boldsymbol{a}_t} \log p_t(\boldsymbol{a}_t|\boldsymbol{s})$ anywhere on the $\boldsymbol{a}_t$ space. □

Proposition 3.1 indicates that the underlying learning target of the score function $s_\theta(\boldsymbol{a}_t; \boldsymbol{s}, t)$ is the *noise-perturbed score functions $\nabla_{\boldsymbol{a}_t} \log p_t(\boldsymbol{a}_t|\boldsymbol{s})$*. At the sampling stage, as the noise gets close to zero when $t$ goes from $T$ to $1$ in the reverse process (8), the noise-perturbed EBMs gradually resemble the original noiseless target data distribution $p_0$, Therefore, diffusion models can be regarded as a series of noise-perturbed EBMs[2], and we can use the diffusion policy to express the energy-based policies such as (2) and (5).

**Revisiting the challenges in online RL.** Proposition 3.1 shows that the DSM loss (9) is a tractable and efficient way to train the score network to match $\nabla_{\boldsymbol{a}_t} \log p_t$ when we have access to samples from $p_0$. However, training diffusion policy is highly non-trivial in online RL because of two major challenges:

• **Sampling challenge:** In online RL, we do not have data samples from the policies such as (2) or (5), the DSM

---

[2]The reason to add noise perturbations in score functions is to encourage exploration on the energy landscape, which significantly improves the sampling quality and makes diffusion-like models the key breakthrough in EBMs (Song & Ermon, 2019).

loss (9) is no longer tractable.

• **Computational challenge:** Another possible solution is to treat the reverse process as policy parameterizations and backpropagate policy gradient through the whole reverse diffusion process (8) like Wang et al. (2024); Celik et al. (2025). However, this recursive gradient propagation not only incurs huge computational and memory costs, making diffusion policy learning expensive and unstable. Moreover, this policy parametrization viewpoint is limited to the max-entropy policy formulation (5).

These challenges hinder the feasibility and performance of diffusion-based policies in online RL. We need a principal way to train diffusion policies when we have the energy function as partial or full knowledge of the data distribution, which we reveal in the following.

### 3.2. Reweighted Score Matching

We develop our core contribution, Reweighted Score Matching (RSM), a general loss family leading to efficient diffusion policy learning algorithms that eliminate the aforementioned difficulties.

A key observation from VSM loss (10) is that, integrating the square error $\|s_\theta(\boldsymbol{a}_t; \boldsymbol{s}, t) - \nabla_{\boldsymbol{a}_t} \log p_t(\boldsymbol{a}_t|\boldsymbol{s})\|^2$ over distribution $p_t(\cdot|\boldsymbol{s})$ is not the only option to perform score matching that matches $s_\theta(\boldsymbol{a}_t, \boldsymbol{s}, t)$ with $\nabla_{\boldsymbol{a}_t} \log p_t(\boldsymbol{a}_t|\boldsymbol{s})$. Our core idea is to generalize this by reweighting the VSM loss (10), allowing integrations with respect to any strictly positive function $g(\boldsymbol{a}_t; \boldsymbol{s}) : \mathcal{A} \times \mathcal{S} \to (0, \infty)$ as long as the following loss function is well-defined,

$$\mathcal{L}^g(\theta; \boldsymbol{s}, t) = \int g(\boldsymbol{a}_t; \boldsymbol{s}) \|s_\theta(\boldsymbol{a}_t; \boldsymbol{s}, t) - \nabla_{\boldsymbol{a}_t} \log p_t(\boldsymbol{a}_t|\boldsymbol{s})\|^2 d\boldsymbol{a}_t \quad (12)$$

where the superscript $g$ indicates the reweighting function. The optimal solution of minimizing $\mathcal{L}^g(\theta)$ remains the same as matching $\nabla_{\boldsymbol{a}_t} \log p_t(\boldsymbol{a}_t|\boldsymbol{s})$ everywhere on $\boldsymbol{a}_t$ space in Proposition 3.1, $s_\theta(\boldsymbol{a}_t; \boldsymbol{s}, t)$. (12) indicates a general loss family, where the VSM loss in (10) lies in it as $\mathcal{L}^{p_t}(\theta; \boldsymbol{s}, t)$.

The reweighting technique gives us more flexibility in loss function design. We will show tractable equivalent formulations in the next section.

## 4. Diffusion Policy Optimization using Reweighted Score Matching

In this section, we show two different reweighting functions, with which the loss $\mathcal{L}^g$ can be converted to tractable loss functions, to train diffusion policies to represent both the mirror descent policy (2) and softmax policy (5). We also discuss practical issues, such as the exploration-exploitation tradeoff, sampling distributions.

### 4.1. Tractable Reweighted Loss Functions

#### 4.1.1. DIFFUSION POLICY MIRROR DESCENT

Consider the mirror descent policy $\pi_{\mathrm{MD}}(\cdot|\boldsymbol{s})$ in (2) and set $p_0(\cdot|\boldsymbol{s}) = \pi_{\mathrm{MD}}(\cdot|\boldsymbol{s})$, we define the reweighting function as

$$g_{\mathrm{MD}} = Z_{\mathrm{MD}}(\boldsymbol{s}) p_t(\boldsymbol{a}_t|\boldsymbol{s})$$

where $Z_{\mathrm{MD}}(\boldsymbol{s}) = \int \pi_{\mathrm{old}}(\boldsymbol{a}|\boldsymbol{s}) \exp\left(Q(\boldsymbol{s}, \boldsymbol{a})/\lambda\right) d\boldsymbol{a}$. Then we can show the reweighted loss $\mathcal{L}^{g_{\mathrm{MD}}}(\theta; \boldsymbol{s}, t)$ is tractable via the following derivation,

$$\mathcal{L}^{g_{\mathrm{MD}}}(\theta; \boldsymbol{s}, t)$$
$$= \int g_{\mathrm{MD}}(\boldsymbol{a}_t; \boldsymbol{s}) \|s_\theta(\boldsymbol{a}_t; \boldsymbol{s}, t) - \nabla_{\boldsymbol{a}_t} \log p_t(\boldsymbol{a}_t|\boldsymbol{s})\|^2 d\boldsymbol{a}_t =$$
$$\underbrace{\mathbb{E}_{\substack{\boldsymbol{a}_0 \sim \pi_{\mathrm{old}} \\ \boldsymbol{a}_t \sim q_{t|0}}} \left[\exp\left(\frac{Q(\boldsymbol{s}, \boldsymbol{a}_0)}{\lambda}\right) \|s_\theta(\boldsymbol{a}_t; \boldsymbol{s}, t) - \nabla_{\boldsymbol{a}_t} \log q_{t|0}(\boldsymbol{a}_t|\boldsymbol{a}_0)\|^2\right]}_{\mathcal{L}_{\mathrm{DPMD}}(\theta, \boldsymbol{s}, t)}$$
$$+ \texttt{constant}$$
$$(13)$$

where $\mathcal{L}_{\mathrm{DPMD}}(\theta, \boldsymbol{s}, t)$ is tractable through unbiased sampling-based approximation. The derivation is similar to Proposition 3.1, and we defer it to Appendix B.2.

#### 4.1.2. SOFT DIFFUSION ACTOR-CRITIC

The max-entropy policy $\pi_{\mathrm{MaxEnt}}$ in (5) is more challenging as we only know the energy function. It is also closely related to the Boltzmann sampling problem (Akhound-Sadegh et al., 2024; Midgley et al., 2022). We need a special sampling protocol to handle it. First, we define the reweighting function as

$$g_{\mathrm{MaxEnt}} = h_t(\boldsymbol{a}_t|\boldsymbol{s}) Z(\boldsymbol{s}) p_t(\boldsymbol{a}_t|\boldsymbol{s})$$

where $h_t(\boldsymbol{a}_t|\boldsymbol{s})$ is a sampling distribution *we choose*. We require $h_t(\boldsymbol{a}_t|\boldsymbol{s})$ to have full support on $\boldsymbol{a}_t$ space. Then we can show the following equivalence,

$$\mathcal{L}^{g_{\mathrm{MaxEnt}}}(\theta; \boldsymbol{s}, t)$$
$$= \int g_{\mathrm{MaxEnt}}(\boldsymbol{a}_t; \boldsymbol{s}) \|s_\theta(\boldsymbol{a}_t; \boldsymbol{s}, t) - \nabla_{\boldsymbol{a}_t} \log p_t(\boldsymbol{a}_t|\boldsymbol{s})\|^2 d\boldsymbol{a}_t$$
$$= \texttt{constant} \times$$
$$\underbrace{\mathbb{E}_{\substack{\boldsymbol{a}_t \sim h_t \\ \tilde{\boldsymbol{a}}_0 \sim \phi_{0|t}}} \left[\exp\left(\frac{Q(\boldsymbol{s}, \tilde{\boldsymbol{a}}_0)}{\lambda}\right) \|s_\theta(\boldsymbol{a}_t; \boldsymbol{s}, t) - \nabla_{\boldsymbol{a}_t} \log \phi_{0|t}(\tilde{\boldsymbol{a}}_0|\boldsymbol{a}_t)\|^2\right]}_{\mathcal{L}_{\mathrm{SDAC}}(\theta, \boldsymbol{s}, t)}$$
$$+ \texttt{constant}$$
$$(14)$$

where $\phi_{0|t}$ is a conditional Gaussian distribution defined as

$$\phi_{0|t}(\tilde{\boldsymbol{a}}_0|\boldsymbol{a}_t) := \mathcal{N}\left(\tilde{\boldsymbol{a}}_0; \frac{1}{\sqrt{\bar{\alpha}_t}}\boldsymbol{a}_t, \frac{1 - \bar{\alpha}_t}{\bar{\alpha}_t}\mathbf{I}\right). \quad (15)$$

The reason we introduce $\phi_{0|t}(\tilde{\boldsymbol{a}}_0|\boldsymbol{a}_t)$ is to use the following reverse sampling trick.

*Remark* 4.1. (Reverse sampling trick.) The density func-

tions of $q_{t|0}$ and $\phi_{0|t}$ are

$$\phi_{0|t}(\boldsymbol{a}_0|\boldsymbol{a}_t) = \left(2\pi\frac{1-\bar{\alpha}_t}{\bar{\alpha}_t}\right)^{-d/2}\exp\left(\frac{(\boldsymbol{a}_t-\bar{\alpha}_t\boldsymbol{a}_0)^2}{-2\left(1-\bar{\alpha}_t\right)}\right),$$

$$q_{t|0}(\boldsymbol{a}_t|\boldsymbol{a}_0) = \left(2\pi\left(1-\bar{\alpha}_t\right)\right)^{-d/2}\exp\left(\frac{(\boldsymbol{a}_t-\bar{\alpha}_t\boldsymbol{a}_0)^2}{-2\left(1-\bar{\alpha}_t\right)}\right)$$

where these two density functions only differ by a constant. We show an abstract example of the reverse sampling trick here. Consider the following integral that is well-defined

$$J(\boldsymbol{s}) := \int h_t(\boldsymbol{a}_t|\boldsymbol{s})p_0(\boldsymbol{a}|\boldsymbol{s})q_{t|0}(\boldsymbol{a}_t|\boldsymbol{a}_0)l(\boldsymbol{a}_t,\boldsymbol{a}_0;\boldsymbol{s})d\boldsymbol{a}_0 d\boldsymbol{a}_t$$
$$= \mathbb{E}_{\boldsymbol{a}_0\sim p_0,\boldsymbol{a}_t\sim q_{t|0}}\left[h(\boldsymbol{a}_t|\boldsymbol{s})l(\boldsymbol{a}_t,\boldsymbol{a}_0;\boldsymbol{s})\right]$$

where $l : \mathcal{A}\times\mathcal{A}\times\mathcal{S}\to\mathbb{R}$ is an integrable function. Notice that we can equivalently compute the integral $J(\boldsymbol{s})$ by another expectation,

$$J(\boldsymbol{s}) \propto \int h_t(\boldsymbol{a}_t|\boldsymbol{s})p_0(\boldsymbol{a}|\boldsymbol{s})\phi_{0|t}(\boldsymbol{a}_0|\boldsymbol{a}_t)l(\boldsymbol{a}_t,\boldsymbol{a}_0;\boldsymbol{s})d\boldsymbol{a}_0 d\boldsymbol{a}_t$$
$$= \mathbb{E}_{\boldsymbol{a}_t\sim h,\boldsymbol{a}_0\sim\phi_{0|t}}\left[p_0(\boldsymbol{a}|\boldsymbol{s})l(\boldsymbol{a}_t,\boldsymbol{a}_0;\boldsymbol{s})\right].$$

This trick helps bypass sampling from $p_0$ and sample from the distribution $h_t$ instead. The detailed derivations are in Appendix B.2.

**Summary.** We can see both loss functions (13) and (14) handles the aforementioned sampling and computational challenges. First, we avoid sampling from the target policy $\pi_{\mathrm{MD}}$ or $\pi_{\mathrm{softmax}}$, and sampling from either the current policy $\pi_{\mathrm{old}}$ or a distribution $h_t$ we can choose. Second, we have a similar computation with denoising score matching (7), avoiding extra computational cost induced by diffusion policy learning. These benefits perfectly echo the difficulties of sampling and computations in applying vanilla diffusion model training to online RL, enabling efficient diffusion policy learning.

*Remark* 4.2 (Broader applications). We emphasize that although we develop RSM with the reweighting techniques for online RL problems, the RSM has its own merit and can be applied to enable diffusion models on any probabilistic modeling problem with known energy functions, such as Boltzmann samplers (Akhound-Sadegh et al., 2024; Midgley et al., 2022). We also show a toy example of Boltzmann sampling in Section 5.1 where we use RSM to train a toy diffusion model to generate samples from a Gaussian mixture distribution with only access to the energy functions.

### 4.2. Practical Issues of Diffusion Policy Training

**Batch action sampling.** Ding et al. (2024a) revealed that diffusion models are too random for efficient exploitation, and proposed to sample a batch of cations and choose the one with the highest $Q$-value as the behavior policy,

$$\boldsymbol{a} = \arg\max_i Q(\boldsymbol{s},\boldsymbol{a}^{(i)}). \tag{16}$$

We leverage this trick in both of our algorithms and add a Gaussian noise whose noise level is automatically tuned

to balance exploration and exploitation, similar with Wang et al. (2024).

**Log likelihood computation.** The soft policy evaluation step in SDAC requires explicit log-likelihood that is non-trivial for diffusion policies. However, we observe that the action after batch action sampling is of low stochasticity, thus, we can use the log probability of the additive Gaussian to approximate the log probability of the policy.

**Numerical stability.** In practice, the exponential of large $Q$ functions in (13) and (14) might cause the loss to explode. We handle the numerical stabilities by **(a) Normalization.** In DPMD, we normalize the $Q(\boldsymbol{s},\boldsymbol{a}_0)$ with the exponential moving average (EMA) of mean and standard deviation over the sampled minibatch. **(b) The `logsumexp` trick.** In SDAC, we sample multiple $\tilde{a}_0^{(i)} \sim \phi_{0|t}, i \in \{1,2,\ldots,K\}$ for every $\boldsymbol{s},\boldsymbol{a}_t$ and use the `logsumexp` trick to avoid explosion of the weights, which means replacing the $\exp(Q(\boldsymbol{s},\tilde{\boldsymbol{a}}_0)/\lambda)$ in (14) with

$$\exp\left(Q(\boldsymbol{s},\tilde{\boldsymbol{a}}_0^{(i)})/\lambda - \log\sum_{(i)}\exp\left(Q(\boldsymbol{s},\tilde{\boldsymbol{a}}_0^{(i)})/\lambda\right)\right),$$

The trick does not conflict with our theoretical derivation, which is another reweighting on the $\boldsymbol{s}$ space.

**Reverse sampling distribution selection.** In SDAC, we can choose the sampling distribution $h_t$. Empirically we tried uniform distribution, last policy $\pi_{\mathrm{old}}(\cdot|\boldsymbol{s})$, and the perturbed data distributions $\int \pi_{\mathrm{old}}(\boldsymbol{a}_0|\boldsymbol{s})q_{t|0}(\cdot|\boldsymbol{a}_0)d\boldsymbol{a}_0$. All these distributions show similar performance.

Combining all the discussions above, we present the practical algorithm of DPMD in Algorithm 1, while SDAC are detailed in Appendix C.2.

### 4.3. Comparison with Recent Diffusion-based Online RLs

We say both proposed algorithms are efficient because of similar computation and memory cost with denoising score matching (7) while bypassing the sampling issues, while recent diffusion-based online RL either incur huge computational or memory cost or induce approximation errors. Recent works on diffusion policy online RLs can be categorized into these families: **i) Score-based Bolzmann sampling.** With the known energy functions in (5), Psenka et al. (2023); Jain et al. (2024) differentiated it to get the non-noisy score function and use Langevin dynamics or diffusion to sample from (5). The empirical performance is not good due to the inaccurate score function obtained by differentiating a learned energy function. **ii) Reverse diffusion as policy parametrizations.** The reverse process (8) can also be directly regarded as a complex parametrization of $\theta$. Wang et al. (2024) backpropagate policy gradients through the reverse diffusion process, resulting in huge com-

---

**Algorithm 1** Diffusion Policy Mirror Descent (DPMD)

---

**Require:** Diffusion noise schedule $\beta_t, \bar{\alpha}_t$ for $t \in \{1, 2, \ldots T\}$, MDP $\mathcal{M}$, initial policy parameters $\theta_0$, initial Q-function parameters $\zeta_0$, replay buffer $\mathcal{D} = \emptyset$, learning rate $\beta$, KL-divergence coefficient $\lambda_0$ and target $\lambda_{\text{target}}$, $\mu_Q(0) = 0.0$, $\sigma_Q(0) := 1.0$, EMA parameter $\xi$

1: **for** epoch $e = 1, 2, \ldots$ **do**
2:     Sample $M$ actions with policy $s_{\theta_{e-1}}$ and choose one according to (16).
3:     Interact with $\mathcal{M}$, store the data in update replay buffer $\mathcal{D}$.
4:     Sample a minibatch of $(\boldsymbol{s}, \boldsymbol{a}, r, \boldsymbol{s}')$ from $\mathcal{D}$.
5:     *# Policy evaluation.*
6:     Sample $\boldsymbol{a}'$ via reverse diffusion process (8) with $s_{\theta_{e-1}}$.
7:     Update $Q_e$ by minimizing the Bellman residual (28).
8:     *# Diffusion Policy Mirror Descent.*
9:     Sample $t$ uniformly from $\{1, 2, \ldots, T\}$. Sample $\boldsymbol{a}_0$ using $s_{\theta_{e-1}}$ and $\boldsymbol{a}_t \sim q_{t|0}(\cdot|\boldsymbol{a}_0)$.
10:     Compute $Q_e(\boldsymbol{s}, \boldsymbol{a}_0)$ and normalize $\bar{Q}_e(\boldsymbol{s}, \boldsymbol{a}_0) = \frac{Q_e(\boldsymbol{s}, \boldsymbol{a}_0) - \mu_Q(e-1)}{\sigma_Q(e-1)}$ update $\theta_e$ with score matching $\mathbb{E}_{\boldsymbol{s},t}\left[\mathcal{L}_{\text{DPMD}}(\theta_{e-1}; \boldsymbol{s}, t)\right]$ in (13) with $Q_e$.
11:     Update KL-divergence coefficient $\lambda_e \leftarrow \lambda_{e-1} + \beta(\lambda_{e-1} - \lambda_{\text{target}})$.
12:     Update EMA $\mu_Q(e) = (1 - \xi)\mu_Q(e-1) + \xi \operatorname{mean}(Q_e)$, $\sigma_Q(e) = (1-\xi)\sigma_Q(e-1) + \xi \operatorname{std}(Q_e)$
13: **end for**

---

putation costs. Ding et al. (2024a) approximate the policy learning as a maximum likelihood estimation for the reverse process, which incurs approximation errors and can not handle negative $Q$-values. **iii) Others.** Yang et al. (2023) maintained a separate diffusion buffer to approximate the policy distribution and fit it with the diffusion model. Ren et al. (2024) combined the reverse process MDP with MDP in RL and conducted policy optimizations. They all induce huge memory and computation costs, thus being impractical and unnecessary. More general related works can be found in Appendix A.

# 5. Experimental Results

This section presents the experimental results. We first use a toy example, generating a 2D Gaussian mixture, to verify the effectiveness of the proposed reweighted score matching as diffusion model training. Then we show the empirical results of the proposed DPMD and SDAC algorithms evaluated with OpenAI Gym MuJoCo tasks.

## 5.1. Toy Example

We first show a toy example of generating a 2D Gaussian mixture distribution from the known energy function (also

known as Boltzmann sampling) to verify the effectiveness of the proposed reweighted score matching. The Gaussian mixture model is composed of two modes whose mean values are $[3, 3]$ and $[-3, -3]$ and mixing coefficients are 0.8 and 0.2 shown in Figure 2(a). The detailed training setup can be found in Appendix C.3, while another Boltzmann sampling task named Two Moon is shown in Appendix C.4.

As we only know the energy function, we select the SDAC-like loss function in (14) as our training objective. We compare three diffusion models trained with two types of loss functions: **a.** proposed SDAC-like loss function in (14) with sampling distribution $h$ being Gaussian and uniform distributions in Figure 2(b) and Figure 2(c), which have access to the true energy function but cannot sample directly from the Gaussian mixture. **c.** Denoising score matching loss (9) in Figure 2(d), which has access to sample from the Gaussian mixture. Empirical results showed that all diffusion models can approximately recover both the two modes and the mixing coefficients, which verifies the effectiveness of the proposed RSM approach to train diffusion models.

Moreover, we also show the naive Langevin dynamics (Neal et al., 2011) samples as a reference in Figure 2(e), which has access to the true score function without noise perturbations. It shows that even with the true score function, Langevin dynamics can not correctly recover the mixing coefficient in finite steps (20 steps in this case), demonstrating the necessity of diffusion models even with given energy functions.

## 5.2. OpenAI Gym MuJoCo Tasks

### 5.2.1. EXPERIMENTAL SETUP

We implemented the proposed DPMD and SDAC algorithms with the JAX package[3] and evaluated the performance on 10 OpenAI Gym MuJoCo v4 tasks. All environments except Humanoid-v4 are trained over 200K iterations with a total of 1 million environment interactions, while Humanoid-v4 has five times more.

**Baselines.** The baselines include two families of model-free RL algorithms. The first family is diffusion policy RL, which includes a collection of recent diffusion-policy online RLs, including QSM (Psenka et al., 2023), QVPO (Ding et al., 2024a), DACER (Wang et al., 2024), DIPO (Yang et al., 2023) and DPPO (Ren et al., 2024). The second family is classic model-free online RL baselines including PPO (Schulman et al., 2017), TD3 (Fujimoto et al., 2018) and SAC (Haarnoja et al., 2018). A more detailed explanation to the baselines can be found in Appendix C.5.

---

[3]The implementation can be found at https://github.com/mahaitongdae/diffusion_policy_online_rl.

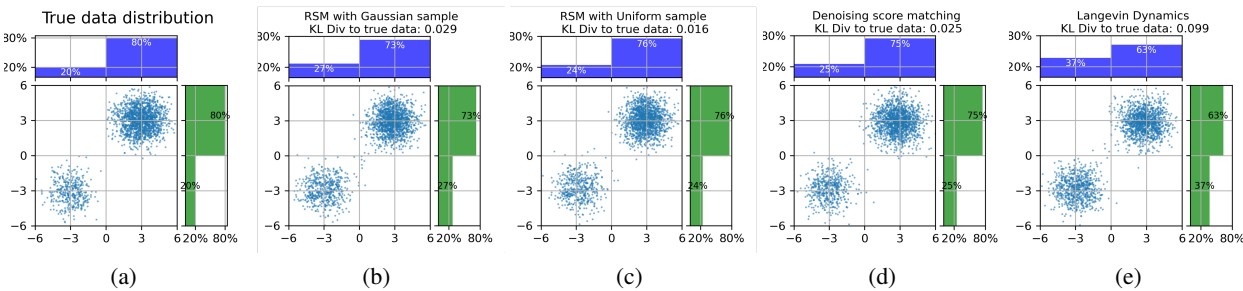

*Figure 2.* The scatter plots of generating 2D Gaussian mixture, the histograms show the partition on each axis. Figure 2(a) shows the true data samples with mixing coefficients [0.8, 0.2]. Figures 2(b) to 2(d) show that the proposed reweighted score matching and denoising score matching can approximately recover the true data distribution. Figure 2(e) shows the slow mixing of Langevin dynamics that the mixing coefficients can not be correctly recovered.

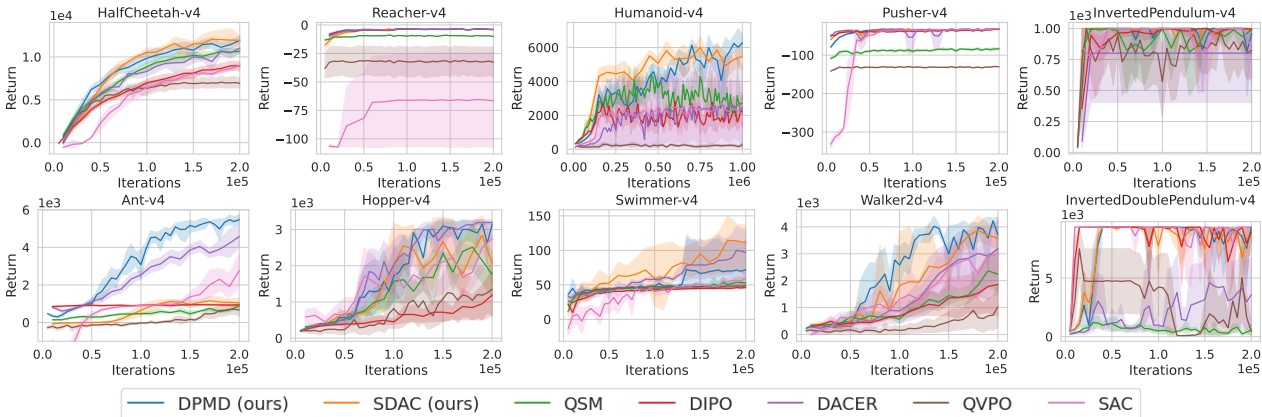

*Figure 3.* Average return over 20 evaluation episodes every 25k iterations (125k for Humanoid) during training. We select the top 5 baselines ranked by average performance over all tasks for clarity. The error bars are standard deviations over 5 random seeds.

### 5.2.2. EXPERIMENTAL RESULTS

The performance and training curves are shown in Table 1 and Figure 3, which shows that our proposed algorithm outperforms all the baselines in all OpenAI Gym MuJoCo environments. Especially, for those complex locomotion tasks including the `HalfCheetah`, `Walker2d`, `Ant`, and `Humanoid`, our top-performing algorithm variant obtained **36.0%, 41.7%, 127.3%, 143.5%** performance improvement compared to SAC. Specifically, `DPMD` achieved **at least 6.4%, 43.4%, 32.1%, 23.1%** performance improvement compared to other diffusion-policy online RL baselines (not the same for all environments), respectively, while `SDAC` shows comparable performance with `DPMD` except `Ant`. The empirical results demonstrate the superior and consistent performance of our proposed algorithm and the true potential of diffusion policies in online RL.

Moreover, the performance of `DPMD` is very stable and consistently good for all the tasks. Other algorithms, including `SDAC`, performed badly on one or some tasks. For example, QSM failed the InvertedDoublePendulum, possibly because its true value function is known to be highly non-smooth. The non-smooth nature results in bad score function esti-

mations since QSM matches the score function by differentiating the $Q$-functions. QVPO failed Reacher and Pusher since it cannot handle negative $Q$-functions. DACER failed InvertedPendulum despite its good performance in some complex tasks, probably due to the gradient instability when backpropagated recursively.

**Computation and memory cost.** We count the GPU memory allocations and total computation time listed in Table 2. The computation is conducted on a desktop workstation with AMD Ryzen 9 7950X CPU, 96 GB memory, and NVIDIA RTX 4090 GPU. We achieve low memory consumption and faster computations compared to other diffusion-policy baselines. Note that the QSM essentially does not involve the denoising process, thus has the lowest computation requirements. We can still achieve a comparable computation time and memory cost with QSM, indicating the proposed `RSM` does not add much extra computational cost due to the diffusion policies. `SDAC` does not need to sample from the current policy to perform reweighted score matching, so it runs faster than `DPMD`.

**Sensitivity analysis.** In Figure 4, we perform sensitivity analyses of different diffusion steps and diffusion noise

*Table 1.* Performance on OpenAI Gym MuJoCo environments. The numbers show the best mean returns and standard deviations over 200k iterations and 5 random seeds.

|  |  | HALFCHEETAH | REACHER | HUMANOID | PUSHER | INVERTEDPENDULUM |
|---|---|---|---|---|---|---|
| **Classic Model-Free RL** | PPO | $4852 \pm 732$ | $-8.69 \pm 11.50$ | $952 \pm 259$ | $-25.52 \pm 2.60$ | $\mathbf{1000 \pm 0}$ |
|  | TD3 | $8149 \pm 688$ | $\mathbf{-3.10 \pm 0.07}$ | $5816 \pm 358$ | $\mathbf{-25.07 \pm 1.01}$ | $\mathbf{1000 \pm 0}$ |
|  | SAC | $8981 \pm 370$ | $-65.35 \pm 56.42$ | $2858 \pm 2637$ | $-31.22 \pm 0.26$ | $\mathbf{1000 \pm 0}$ |
| **Diffusion Policy RL** | QSM | $10740 \pm 444$ | $-4.16 \pm 0.28$ | $5652 \pm 435$ | $-80.78 \pm 2.20$ | $\mathbf{1000 \pm 0}$ |
|  | DIPO | $9063 \pm 654$ | $-3.29 \pm 0.03$ | $4880 \pm 1072$ | $-32.89 \pm 0.34$ | $\mathbf{1000 \pm 0}$ |
|  | DACER | $11203 \pm 246$ | $-3.31 \pm 0.07$ | $2755 \pm 3599$ | $-30.82 \pm 0.13$ | $801 \pm 446$ |
|  | QVPO | $7321 \pm 1087$ | $-30.59 \pm 16.57$ | $421 \pm 75$ | $-129.06 \pm 0.96$ | $\mathbf{1000 \pm 0}$ |
|  | DPPO | $1173 \pm 392$ | $-6.62 \pm 1.70$ | $484 \pm 64$ | $-89.31 \pm 17.32$ | $\mathbf{1000 \pm 0}$ |
|  | **DPMD** | $11924 \pm 609$ | $\mathbf{-3.14 \pm 0.10}$ | $\mathbf{6959 \pm 460}$ | $\mathbf{-30.43 \pm 0.37}$ | $\mathbf{1000 \pm 0}$ |
|  | **SDAC** | $\mathbf{12210 \pm 964}$ | $-3.37 \pm 0.42$ | $6437 \pm 177$ | $-32.53 \pm 5.27$ | $\mathbf{1000 \pm 0}$ |
|  |  | ANT | HOPPER | SWIMMER | WALKER2D | INVERTED2PENDULUM |
| **Classic Model-Free RL** | PPO | $3442 \pm 851$ | $3227 \pm 164$ | $84.5 \pm 12.4$ | $4114 \pm 806$ | $9358 \pm 1$ |
|  | TD3 | $3733 \pm 1336$ | $1934 \pm 1079$ | $71.9 \pm 15.3$ | $2476 \pm 1357$ | $\mathbf{9360 \pm 0}$ |
|  | SAC | $2500 \pm 767$ | $3197 \pm 294$ | $63.5 \pm 10.2$ | $3233 \pm 871$ | $9359 \pm 1$ |
| **Diffusion Policy RL** | QSM | $938 \pm 164$ | $2804 \pm 466$ | $57.0 \pm 7.7$ | $2523 \pm 872$ | $2186 \pm 234$ |
|  | DIPO | $965 \pm 9$ | $1191 \pm 770$ | $46.7 \pm 2.9$ | $1961 \pm 1509$ | $9352 \pm 3$ |
|  | DACER | $4301 \pm 524$ | $3212 \pm 86$ | $103.0 \pm 45.8$ | $3194 \pm 1822$ | $6289 \pm 3977$ |
|  | QVPO | $718 \pm 336$ | $2873 \pm 607$ | $53.4 \pm 5.0$ | $2337 \pm 1215$ | $7603 \pm 3910$ |
|  | DPPO | $60 \pm 15$ | $2175 \pm 556$ | $106.1 \pm 6.5$ | $1130 \pm 686$ | $9346 \pm 4$ |
|  | **DPMD** | $\mathbf{5683 \pm 138}$ | $\mathbf{3275 \pm 55}$ | $79.3 \pm 52.5$ | $\mathbf{4365 \pm 266}$ | $\mathbf{9360 \pm 0}$ |
|  | **SDAC** | $1391 \pm 202$ | $2955 \pm 370$ | $\mathbf{119.1 \pm 41.9}$ | $3995 \pm 498$ | $\mathbf{9360 \pm 0}$ |

*Table 2.* GPU memory allocation and total compute time of 200K iterations and 1 million environment interactions. *QSM did not learn diffusion policies essentially thus the computation is lightweight.

| Algorithm | GPU Memory (MB) | Training time (min) |
|---|---|---|
| QSM* | 997 | 14.23 |
| QVPO | 5219 | 30.90 |
| DACER | 1371 | 27.61 |
| DIPO | 5096 | 19.31 |
| DPPO | 1557 | 95.16 |
| DPMD | 1192 | 21.10 |
| **SDAC** | **1113** | **16.10** |

schedules on the DPMD variant. Results show that 10 and 20 diffusion steps obtain comparable results, both outperforming the 30-step setting. The linear and cosine noise schedules perform similarly, and both outperform the variance-preserving schedule. Therefore, we choose 20 steps and cosine schedules for all tasks. The results also show the robustness to the diffusion process hyperparameters.

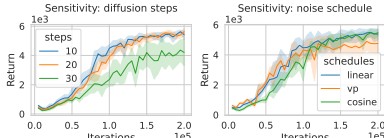

*Figure 4.* Sensitivity analysis on diffusion steps and diffusion noise schedule on Ant-v4.

## 6. Conclusion

In this paper, we proposed Reweighted Score Matching (RSM), an efficient diffusion policy training algorithm tailored for online RL. Regarding diffusion models as noise-perturbed EBMs, we develop the reweighted score matching to train diffusion models with access only to the energy functions and bypass sampling from the data distribution. In this way, we can train a diffusion policy with only access to the $Q$-function as the energy functions in online RL. Empirical results have shown superior performance compared to SAC and other recent diffusion policy online RLs. Possible future directions include improving the stability of diffusion policies and efficient exploration scheme design.

## Impact Statement

This paper presents work whose goal is to advance the field of Machine Learning. There are many potential societal consequences of our work, none of which we feel must be specifically highlighted here.

## Acknowledgment

This paper is supported by NSF AI institute: 2112085, NSF ECCS-2401390, NSF ECCS-2401391, ONR N000142512173, NSF ASCENT 2328241, NSF IIS-2403240, Schmidt Sciences AI2050 Fellowship.

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

# A. Related works

**Diffusion models for decision making.** Due to their rich expressiveness in modeling complex and multimodal distributions, diffusion models have been leveraged to represent stochastic policies (Wang et al., 2022; Chen et al., 2022; Hansen-Estruch et al., 2023), plan trajectories (Janner et al., 2022; Chi et al., 2023; Du et al., 2024) and capture transition dynamics (Rigter et al., 2023; Ding et al., 2024b; Shribak et al., 2024). Specifically, we focus on the diffusion policies. Diffusion policies have been primarily used on offline RL with expert datasets, where the denoising score matching (7) is still available and the learned $Q$-function only provides extra guidance such as regularization (Wang et al., 2022) or multiplication in the energy function. However, in online RL we do not have the dataset, thus denoising score matching is impossible.

**Diffusion Models.** Diffusion models have a dual interpretation of EBMs and latent variable models. The latent variable interpretation is motivated by the solving reverse-time diffusion thermodynamics via multiple layers of decoder networks (Sohl-Dickstein et al., 2015). It was later refined by Ho et al. (2020) via simplified training loss. The EBM interpretation aims to solve pitfalls in Langevin dynamics sampling by adding progressively decreasing noise (Song & Ermon, 2019). Then the two viewpoints are merged together with viewpoints from stochastic differential equations (Song et al., 2021), followed by numerous improvements on the training and sampling design (Song et al., 2022; Karras et al., 2022).

**Noise-conditioned score networks.** A equivalent approaches developed by Song & Ermon (2019) simultaneously with diffusion models is to fit the score function of a series of noise-perturbed data distribution $\mathcal{N}\left(\mathbf{x}_i; \mathbf{x}, \sigma_i^2 \mathbf{I}\right), i = \{1, 2, \ldots, K\}$ with a noise schedule $\sigma_1 > \sigma_2 > \cdots > \sigma_K$. The resulting models, named the noise-conditioned score networks (NCSN) $f_\theta\left(\mathbf{x}_i; \sigma_i\right)$, take the noise level into the inputs and are learned by denoising score matching (Vincent, 2011)

$$\mathbb{E}_{\mathbf{x} \sim p, \mathbf{x}_i \sim \mathcal{N}\left(\mathbf{x}, \sigma_i^2 \mathbf{I}\right)}\left[\left\|f_\theta\left(\mathbf{x}_i; \sigma_i\right) - \nabla_{\mathbf{x}_i} \log q(\mathbf{x}_i | \mathbf{x})\right\|^2\right] \tag{17}$$

Then in the sampling stage, Song & Ermon (2019) uses the Langevin dynamics $\mathbf{x}_{i+1} = \mathbf{x}_i + \eta \nabla_{\mathbf{x}_i} \log q(\mathbf{x}_i \mid \mathbf{x}) + \sqrt{2\eta}\mathbf{z}_i$ to sample from energy function. Song & Ermon (2019) additionally replace the original score function $\nabla_{\mathbf{x}_i} \log q(\mathbf{x}_i \mid \mathbf{x})$ in the Langevin dynamics with the learned noisy score function $f_\theta(\tilde{\mathbf{x}}; \sigma_i)$:

$$\mathbf{x}_{i+1} \leftarrow \mathbf{x}_i + \eta f_\theta(\tilde{\mathbf{x}}; \sigma_i) + \sqrt{2\eta}\mathbf{z}_i, \quad i = 0, \cdots, K \tag{18}$$

named as annealed Langevin dynamics. The scheduled noise perturbation design significantly improved the image generation performance to match the state-of-the-art (SOTA) at that time (Song & Ermon, 2019), which is further refined by DDPM.

We can see that the annealed Langevin dynamics (18) resembles the DDPM sampling (8) with different scale factors, and the denoising score matching loss (9) is equivalent to (7). Therefore, DDPM can be interpreted as EBMs with multi-level noise perturbations. A more thorough discussion on their equivalency can also be found in (Ho et al., 2020; Song et al., 2021).

# B. Derivations

## B.1. Derivations of Proposition 3.1

We repeat Proposition 3.1 here,

**Proposition B.1** (Diffusion models as noise-perturbed EBMs)**.** *Consider a single term in loss function of DDPM* (7) *at given state $\boldsymbol{s}$ and time $t$,*

$$\mathcal{L}_{\mathrm{DSM}}(\theta; \boldsymbol{s}, t) := \\ \mathbb{E}_{\substack{\boldsymbol{a}_0 \sim p_0(\cdot | s) \\ \boldsymbol{a}_t \sim q_{t|0}(\cdot | \boldsymbol{a}_0)}}\left[\left\|s_\theta\left(\boldsymbol{a}_t; \boldsymbol{s}, t\right) - \nabla_{\mathbf{x}_t} \log q_{t|0}(\boldsymbol{a}_t | \boldsymbol{a}_0)\right\|^2\right] \tag{9}$$

*We refer to it as the denoising score matching (DSM) loss. The optimal solution $\theta^*$ is achieved when the following holds for all $\boldsymbol{a}_t$*

$$s_{\theta^*}(\boldsymbol{a}_t; \boldsymbol{s}, t) = \nabla_{\boldsymbol{a}_t} \log p_t(\boldsymbol{a}_t | \boldsymbol{s})$$

*where $p_t(\boldsymbol{a}_t | \boldsymbol{s}) = \int q_{t|0}(\boldsymbol{a}_t | \boldsymbol{a}_0) p_0(\boldsymbol{a}_0 | \boldsymbol{s}) d\boldsymbol{a}_0$ is the noise-perturbed policy with perturbation kernel $q_{t|0}(\boldsymbol{a}_t | \boldsymbol{a}_0) = \mathcal{N}(\boldsymbol{a}_t; \sqrt{\bar{\alpha}_t}\boldsymbol{a}_0, (1 - \bar{\alpha}_t)\mathbf{I})$, for noise schedule index $t = 1, 2, \ldots, T$.*

*Proof.* We first check the square error $\|s_\theta(\boldsymbol{a}_t, \boldsymbol{s}, t) - \nabla_{\boldsymbol{a}_t} \log p_t(\boldsymbol{a}_t | \boldsymbol{s})\|^2$,

$$\|s_\theta(\boldsymbol{a}_t, \boldsymbol{s}, t) - \nabla_{\boldsymbol{a}_t} \log p_t(\boldsymbol{a}_t|\boldsymbol{s})\|^2$$

$$= \left\|s_\theta(\boldsymbol{a}_t, \boldsymbol{s}, t) - \frac{1}{p_t(\boldsymbol{a}_t|\boldsymbol{s})} \nabla_{\boldsymbol{a}_t} \int q_{t|0}(\boldsymbol{a}_t|\boldsymbol{a}_0) p_0(\boldsymbol{a}_0|\boldsymbol{s}) d\boldsymbol{a}_0 \right\|^2$$

$$= \|s_\theta(\boldsymbol{a}_t, \boldsymbol{s}, t)\|^2$$

$$- \frac{1}{p_t(\boldsymbol{a}_t|\boldsymbol{s})} \left\langle s_\theta(\boldsymbol{a}_t, \boldsymbol{s}, t), \int \nabla_{\boldsymbol{a}_t} \log q_{t|0}(\boldsymbol{a}_t|\boldsymbol{a}_0) q_{t|0}(\boldsymbol{a}_t|\boldsymbol{a}_0) p_0(\boldsymbol{a}_0|\boldsymbol{s}) d\boldsymbol{a}_0 \right\rangle + \texttt{constant} \tag{19}$$

$$= \|s_\theta(\boldsymbol{a}_t, \boldsymbol{s}, t)\|^2 \underbrace{\frac{1}{p_t(\boldsymbol{a}_t|\boldsymbol{s})} \int q_{t|0}(\boldsymbol{a}_t|\boldsymbol{a}_0) p_0(\boldsymbol{a}_0|\boldsymbol{s}) d\boldsymbol{a}_0}_{1}$$

$$- \frac{1}{p_t(\boldsymbol{a}_t|\boldsymbol{s})} \int q_{t|0}(\boldsymbol{a}_t|\boldsymbol{a}_0) p_0(\boldsymbol{a}_0|\boldsymbol{s}) \left\langle s_\theta(\boldsymbol{a}_t, \boldsymbol{s}, t), \nabla_{\boldsymbol{a}_t} \log q_{t|0}(\boldsymbol{a}_t|\boldsymbol{a}_0) \right\rangle d\boldsymbol{a}_0 + \texttt{constant}$$

$$= \frac{1}{p_t(\boldsymbol{a}_t|\boldsymbol{s})} \int q_{t|0}(\boldsymbol{a}_t|\boldsymbol{a}_0) p_0(\boldsymbol{a}_0|\boldsymbol{s}) \left( \|s_\theta(\boldsymbol{a}_t, \boldsymbol{s}, t)\|^2 - 2 \left\langle s_\theta(\boldsymbol{a}_t, \boldsymbol{s}, t), \nabla_{\boldsymbol{a}_t} \log q_{t|0}(\boldsymbol{a}_t|\boldsymbol{a}_0) \right\rangle \right) d\boldsymbol{a}_0 + \texttt{constant}$$

$$= \frac{1}{p_t(\boldsymbol{a}_t|\boldsymbol{s})} \int q_{t|0}(\boldsymbol{a}_t|\boldsymbol{a}_0) p_0(\boldsymbol{a}_0|\boldsymbol{s}) \left\| s_\theta(\boldsymbol{a}_t, \boldsymbol{s}, t) - \nabla_{\boldsymbol{a}_t} \log q_{t|0}(\boldsymbol{a}_t|\boldsymbol{a}_0) \right\|^2 d\boldsymbol{a}_0 + \texttt{constant} \tag{20}$$

where the `constant` denotes constants that are irrelevant with $\theta$. (20) builds the foundation of transferring the intractable vanilla score matching loss (10) to tractable denoising score matching that can be easily computed by the sampled Gaussian noise.

Therefore, we can integrate both sides on $p_t(\boldsymbol{a}_t|\boldsymbol{s})$,

$$\mathop{\mathbb{E}}_{\substack{\boldsymbol{a}_0 \sim p_0 \\ \boldsymbol{a}_t \sim q_{t|0}}} \left[ \left\| s_\theta(\boldsymbol{a}_t; \boldsymbol{s}, t) - \nabla_{\boldsymbol{a}_t} \log q_{t|0}(\boldsymbol{a}_t|\boldsymbol{a}_0) \right\|^2 \right] = \mathbb{E}_{\boldsymbol{a}_t \sim p_t} \left[ \left\| s_\theta(\boldsymbol{a}_t; \boldsymbol{s}, t) - \nabla_{\boldsymbol{a}_t} \log p_t(\boldsymbol{a}_t|\boldsymbol{s}_t) \right\|^2 \right] + \texttt{constant}$$

where the LHS is $\mathcal{L}_{\mathrm{DSM}}(\theta; \boldsymbol{s}, t)$ and RHS is $\mathcal{L}_{\mathrm{VSM}}(\theta; \boldsymbol{s}, t) + \texttt{constant}$ defined in (10). This concludes the proof of Proposition 3.1. $\qquad\square$

## B.2. Derivations of Section 4.1

Section 4.1 shows that we can match the score network $s_\theta(\boldsymbol{a}_t; \boldsymbol{s}, t)$ with noise-perturbed policy score function $\nabla_{\boldsymbol{a}_t} \log p_t(\boldsymbol{a}_t|\boldsymbol{s})$ without sampling from $p_0$ like denoising score matching (7).

### B.2.1. DERIVATIONS OF DIFFUSION POLICY MIRROR DESCENT

First, we restate the results. Consider the mirror descent policy $\pi_{\mathrm{MD}}(\cdot|\boldsymbol{s})$ in (2) and set $p_0(\cdot|\boldsymbol{s}) = \pi_{\mathrm{MD}}(\cdot|\boldsymbol{s})$, we define the reweighting function as

$$g_{\mathrm{MD}} = Z_{\mathrm{MD}}(\boldsymbol{s}) p_t(\boldsymbol{a}_t|\boldsymbol{s})$$

where $Z(\boldsymbol{s}) = \int \pi_{\mathrm{old}}(\boldsymbol{a}|\boldsymbol{s}) \exp\left(Q(\boldsymbol{s}, \boldsymbol{a})/\lambda\right) d\boldsymbol{a}$. Then we can show the reweighted loss $\mathcal{L}^{g_{\mathrm{MD}}}(\theta; \boldsymbol{s}, t)$ is tractable the following derivation,

$$\mathcal{L}^{g_{\mathrm{MD}}}(\theta; \boldsymbol{s}, t) = \int g_{\mathrm{MD}}(\boldsymbol{a}_t; \boldsymbol{s}) \|s_\theta(\boldsymbol{a}_t; \boldsymbol{s}, t) - \nabla_{\boldsymbol{a}_t} \log p_t(\boldsymbol{a}_t|\boldsymbol{s})\|^2 d\boldsymbol{a}_t$$

$$= \underbrace{\mathop{\mathbb{E}}_{\substack{\boldsymbol{a}_0 \sim \pi_{\mathrm{old}} \\ \boldsymbol{a}_t \sim q_{t|0}}} \left[ \exp\left(Q(\boldsymbol{s}, \boldsymbol{a}_0)/\lambda\right) \left\| s_\theta(\boldsymbol{a}_t; \boldsymbol{s}, t) - \nabla_{\boldsymbol{a}_t} \log q_{t|0}(\boldsymbol{a}_t|\boldsymbol{a}_0) \right\|^2 \right]}_{\mathcal{L}_{\mathrm{DPMD}}(\theta, \boldsymbol{s}, t)} + \texttt{constant} \tag{13}$$

where $\mathcal{L}_{\mathrm{DPMD}}(\theta, \boldsymbol{s}, t)$ is tractable through sampling-based approximation.

*Proof.* (13)

$$\mathcal{L}^{g_{\text{MD}}}(\theta; \boldsymbol{s}, t)$$

$$= \int g_{\text{MD}}(\boldsymbol{a}_t; \boldsymbol{s}) \left\| s_\theta(\boldsymbol{a}_t; \boldsymbol{s}, t) - \nabla_{\boldsymbol{a}_t} \log p_t(\boldsymbol{a}_t|\boldsymbol{s}) \right\|^2 d\boldsymbol{a}_t$$

Sbustute in (20),

$$= \int g_{\text{MD}}(\boldsymbol{a}_t; \boldsymbol{s}) \frac{1}{p_t(\boldsymbol{a}_t|\boldsymbol{s})} \int q_{t|0}(\boldsymbol{a}_t|\boldsymbol{a}_0) p_0(\boldsymbol{a}_0|\boldsymbol{s}) \left\| s_\theta(\boldsymbol{a}_t, \boldsymbol{s}, t) - \nabla_{\boldsymbol{a}_t} \log q_{t|0}(\boldsymbol{a}_t|\boldsymbol{a}_0) \right\|^2 d\boldsymbol{a}_0 d\boldsymbol{a}_t + \texttt{constant}$$

$$= \int Z_{\text{MD}}(\boldsymbol{s}) \int q_{t|0}(\boldsymbol{a}_t|\boldsymbol{a}_0) \frac{\pi_{\text{old}}(\boldsymbol{a}|\boldsymbol{s}) \exp(Q(\boldsymbol{s}, \boldsymbol{a})/\lambda)}{Z_{\text{MD}}(\boldsymbol{s})} \left\| s_\theta(\boldsymbol{a}_t, \boldsymbol{s}, t) - \nabla_{\boldsymbol{a}_t} \log q_{t|0}(\boldsymbol{a}_t|\boldsymbol{a}_0) \right\|^2 d\boldsymbol{a}_0 d\boldsymbol{a}_t + \texttt{constant}$$

$$= \iint q_{t|0}(\boldsymbol{a}_t|\boldsymbol{a}_0) \pi_{\text{old}}(\boldsymbol{a}|\boldsymbol{s}) \exp(Q(\boldsymbol{s}, \boldsymbol{a})/\lambda) \left\| s_\theta(\boldsymbol{a}_t, \boldsymbol{s}, t) - \nabla_{\boldsymbol{a}_t} \log q_{t|0}(\boldsymbol{a}_t|\boldsymbol{a}_0) \right\|^2 d\boldsymbol{a}_0 d\boldsymbol{a}_t + \texttt{constant}$$

$$= \mathop{\mathbb{E}}_{\substack{\boldsymbol{a}_0 \sim \pi_{\text{old}} \\ \boldsymbol{a}_t \sim q_{t|0}}} \left[ \exp\left(Q\left(\boldsymbol{s}, \boldsymbol{a}_0\right)/\lambda\right) \left\| s_\theta\left(\boldsymbol{a}_t; \boldsymbol{s}, t\right) - \nabla_{\boldsymbol{a}_t} \log q_{t|0}\left(\boldsymbol{a}_t | \boldsymbol{a}_0\right) \right\|^2 \right] + \texttt{constant}$$

$$\qquad \square$$

### B.2.2. Derivations of Soft Diffusion Actor-Critic

We first restate the results. First, we define the reweighting function as

$$g_{\text{softmax}} = h_t(\boldsymbol{a}_t|\boldsymbol{s}) Z(\boldsymbol{s}) p_t(\boldsymbol{a}_t|\boldsymbol{s})$$

where $h_t(\boldsymbol{a}_t|\boldsymbol{s})$ is a sampling distribution *we choose*. We require $h_t(\boldsymbol{a}_t|\boldsymbol{s})$ to have full support on $\boldsymbol{a}_t$ space. Then we can show the following equivalence,

$$\mathcal{L}^{g_{\text{MaxEnt}}}(\theta; \boldsymbol{s}, t) = \int g_{\text{MD}}(\boldsymbol{a}_t; \boldsymbol{s}) \left\| s_\theta(\boldsymbol{a}_t; \boldsymbol{s}, t) - \nabla_{\boldsymbol{a}_t} \log p_t(\boldsymbol{a}_t|\boldsymbol{s}) \right\|^2 d\boldsymbol{a}_t$$

$$= \texttt{constant} \times \underbrace{\mathop{\mathbb{E}}_{\substack{\boldsymbol{a}_t \sim h_t \\ \tilde{\boldsymbol{a}}_0 \sim \phi_{0|t}}} \left[ \exp\left(Q\left(\boldsymbol{s}, \tilde{\boldsymbol{a}}_0\right)/\lambda\right) \left\| s_\theta\left(\boldsymbol{a}_t; \boldsymbol{s}, t\right) - \nabla_{\boldsymbol{a}_t} \log \phi_{0|t}\left(\tilde{\boldsymbol{a}}_0 | \boldsymbol{a}_t\right) \right\|^2 \right]}_{\mathcal{L}_{\text{SDAC}}(\theta, \boldsymbol{s}, t)} + \texttt{constant}$$

$$(14)$$

where $\phi_{0|t}$ is a conditional Gaussian distribution defined as

$$\phi_{0|t}(\tilde{\boldsymbol{a}}_0|\boldsymbol{a}_t) := \mathcal{N}\left(\tilde{\boldsymbol{a}}_0; \frac{1}{\sqrt{\bar{\alpha}_t}} \boldsymbol{a}_t, \frac{1 - \bar{\alpha}_t}{\bar{\alpha}_t} \mathbf{I}\right) \qquad (15)$$

**Proof.** First, we substitute (20) into (14) to get that

$$\mathcal{L}^{g_{\text{MaxEnt}}}(\theta; \boldsymbol{s}, t) = \int h_t(\boldsymbol{a}_t|\boldsymbol{s}) Z(\boldsymbol{s}) p_t(\boldsymbol{a}_t|\boldsymbol{s}) \left\| s_\theta(\boldsymbol{a}_t; \boldsymbol{s}, t) - \nabla_{\boldsymbol{a}_t} \log p_t(\boldsymbol{a}_t|\boldsymbol{s}) \right\|^2 d\boldsymbol{a}_t$$

Sbustute in (20),

$$= \iint h_t(\boldsymbol{a}_t|\boldsymbol{s}) Z(\boldsymbol{s}) p_0(\boldsymbol{a}_0|\boldsymbol{s}) q_{t|0}(\boldsymbol{a}_t|\boldsymbol{a}_0) \left\| s_\theta\left(\boldsymbol{a}_t; \boldsymbol{s}, t\right) - \nabla_{\boldsymbol{a}_t} \log q_{t|0}\left(\boldsymbol{a}_t | \boldsymbol{a}_0\right) \right\|^2 d\boldsymbol{a}_0 d\boldsymbol{a}_t + \texttt{constant} \qquad (21)$$

Then we leverage the reverse sampling trick,

There exists a reverse sampling distribution $\phi_{0|t}$ satisfying

$$\phi_{0|t}(\boldsymbol{a}_0 | \boldsymbol{a}_t) = \mathcal{N}\left(\boldsymbol{a}_0; \frac{1}{\sqrt{\bar{\alpha}_t}} \boldsymbol{a}_t, \frac{1 - \bar{\alpha}_t}{\bar{\alpha}_t} \mathbf{I}\right) \propto q_{t|0}(\boldsymbol{a}_t | \boldsymbol{a}_0) = \mathcal{N}\left(\boldsymbol{a}_t; \sqrt{\bar{\alpha}_t} \boldsymbol{a}_0, (1 - \bar{\alpha}_t) \mathbf{I}\right), \qquad (22)$$

and their score functions match

$$\nabla_{\boldsymbol{a}_t} \log q_{t|0}(\boldsymbol{a}_t | \boldsymbol{a}_0) = \nabla_{\boldsymbol{a}_t} \log \phi_{0|t}(\boldsymbol{a}_0 | \boldsymbol{a}_t) = -\frac{\boldsymbol{a}_t - \sqrt{\bar{\alpha}_t} \boldsymbol{a}_0}{1 - \bar{\alpha}_t} \qquad (23)$$

This is achieved by examining the density function,

$$\phi_{0|t}\left(\tilde{\boldsymbol{a}}_0 \mid \boldsymbol{a}_t\right) = \left(2\pi \frac{1-\bar{\alpha}_t}{\bar{\alpha}_t}\right)^{-d/2} \exp\left(-\frac{\left\|\tilde{\boldsymbol{a}}_0 - \frac{1}{\sqrt{\bar{\alpha}_t}}\boldsymbol{a}_t\right\|^2}{2\frac{(1-\bar{\alpha}_t)}{\bar{\alpha}_t}}\right) \tag{24}$$

while

$$q_{t|0}\left(\boldsymbol{a}_t \mid \boldsymbol{a}_0\right) = (2\pi(1-\bar{\alpha}_t))^{-d/2} \exp\left(-\frac{\|\sqrt{\bar{\alpha}_t}\boldsymbol{a}_0 - \boldsymbol{a}_t\|^2}{2\left(1-\bar{\alpha}_t\right)}\right) = (\bar{\alpha}_t)^{-d/2}\phi_{0|t}\left(\boldsymbol{a}_0 \mid \boldsymbol{a}_t\right) \tag{25}$$

Leveraging the reverse sampling trick, we can change the weighting function in (21) to

$$h_t(\boldsymbol{a}_t|\boldsymbol{s})Z(\boldsymbol{s})p_0(\boldsymbol{a}_0|\boldsymbol{s})q_{t|0}(\boldsymbol{a}_t|\boldsymbol{a}_0)$$
$$= h_t(\boldsymbol{a}_t|\boldsymbol{s})Z(\boldsymbol{s})\frac{\exp(Q(\boldsymbol{s},\boldsymbol{a}_0)/\lambda)}{Z(\boldsymbol{s})}q_{t|0}(\boldsymbol{a}_t|\boldsymbol{a}_0) \tag{26}$$
$$= (\bar{\alpha}_t)^{d/2}\exp(Q(\boldsymbol{s},\boldsymbol{a}_0)/\lambda)h_t(\boldsymbol{a}_t|\boldsymbol{s})\phi_{0|t}(\boldsymbol{a}_0|\boldsymbol{a}_t)$$

Considering the score equivalence, (21) can be further derived to

$$\mathcal{L}^{g_{\text{softmax}}}(\theta)$$
$$= (\bar{\alpha}_t)^{d/2}\iint h_t(\boldsymbol{a}_t|\boldsymbol{s})\phi_{0|t}(\tilde{\boldsymbol{a}}_0|\boldsymbol{a}_t)\exp\left(Q(\boldsymbol{s},\tilde{\boldsymbol{a}}_0)/\lambda\right)\left\|s_\theta\left(\boldsymbol{a}_t;\boldsymbol{s},t\right) - \nabla_{\boldsymbol{a}_t}\log\phi_{0|t}\left(\tilde{\boldsymbol{a}}_0 \mid \boldsymbol{a}_t\right)\right\|^2 d\tilde{\boldsymbol{a}}_0 d\boldsymbol{a}_t + \texttt{constant}$$
$$= (\bar{\alpha}_t)^{d/2}\underbrace{\underset{\substack{\boldsymbol{a}_t \sim h_t \\ \tilde{\boldsymbol{a}}_0 \sim \phi_{0|t}}}{\mathbb{E}}\left[\exp\left(Q\left(\boldsymbol{s},\tilde{\boldsymbol{a}}_0\right)/\lambda\right)\left\|s_\theta\left(\boldsymbol{a}_t;\boldsymbol{s},t\right) - \nabla_{\boldsymbol{a}_t}\log\phi_{0|t}\left(\tilde{\boldsymbol{a}}_0 \mid \boldsymbol{a}_t\right)\right\|^2\right]}_{\mathcal{L}_{\text{SDAC}}(\theta,\boldsymbol{s},t)} + \texttt{constant}$$

$$\tag{27}$$

.

## C. Experiments

### C.1. Policy Evaluation for Entropy-Regularized MDPs

#### C.1.1. POLICY EVALUATION FOR DPMD

The policy evaluation of DPMD minimizes the Bellman residual to learn the $Q$-function parameters,

$$\mathcal{L}^Q\left(\zeta;\pi\right) = \mathbb{E}_{\boldsymbol{s},\boldsymbol{a}\sim\mathcal{D}}\left[\left(Q_\zeta(\boldsymbol{s},\boldsymbol{a}) - \left(r(\boldsymbol{s},\boldsymbol{a}) + \gamma\mathbb{E}_{\boldsymbol{s}'}\mathbb{E}_{\boldsymbol{a}'\sim\pi(\boldsymbol{a}'|\boldsymbol{s}')}\left[Q_{\bar{\zeta}}\left(\boldsymbol{s}',\boldsymbol{a}'\right)\right]\right)\right)^2\right]. \tag{28}$$

#### C.1.2. POLICY EVALUATION FOR SDAC

Following the soft policy evaluation (4), we learn the $Q$-function parameters with the Bellman residual

$$\mathcal{L}^Q\left(\zeta;\pi\right) = \mathbb{E}_{\boldsymbol{s},\boldsymbol{a}\sim\mathcal{D}}\left[\left(Q_\zeta(\boldsymbol{s},\boldsymbol{a}) - \left(r(\boldsymbol{s},\boldsymbol{a}) - \lambda\log\pi\left(\boldsymbol{a}|\boldsymbol{s}\right) + \gamma\mathbb{E}_{\boldsymbol{s}'}\mathbb{E}_{\boldsymbol{a}'\sim\pi(\boldsymbol{a}'|\boldsymbol{s}')}\left[Q_{\bar{\zeta}}\left(\boldsymbol{s}',\boldsymbol{a}'\right)\right]\right)\right)^2\right]. \tag{29}$$

In practice, as we sample a batch of actions and select the one with the highest $Q$-value like (16) and add a Gaussian noise with tunable standard deviations, we directly select the log probability of additive noise as $\log\pi\left(\boldsymbol{a}|\boldsymbol{s}\right)$.

### C.2. Algorithm Details

We show the detailed pseudocode of SDAC here in Algorithm 2.

### C.3. Training Setups for the Toy Example

Consider Gaussian mixture model with density function

$$p_0(\mathbf{x}) = 0.8 * \frac{1}{2\pi}\exp\left(-\frac{\|\mathbf{x} - [3;3]\|^2}{2}\right) + 0.2 * \frac{1}{2\pi}\exp\left(-\frac{\|\mathbf{x} + [3;3]\|^2}{2}\right) \tag{30}$$

---

**Algorithm 2** Soft Diffusion Actor-Critic (`SDAC`)

---

**Require:** Diffusion noise schedule $\beta_t, \bar{\alpha}_t$ for $t \in \{1, 2, \ldots T\}$, MDP $\mathcal{M}$, initial policy parameters $\theta_0$, initial Q-function parameters $\zeta_0$, replay buffer $\mathcal{D} = \emptyset$, learning rate $\beta$, KL-divergence coefficient $\lambda_0$ and target $\lambda_{\text{target}}$

1: **for** epoch $e = 1, 2, \ldots$ **do**
2:     *# Sampling and experience replay.*
3:     Interact with $\mathcal{M}$ using policy $s_{\theta_{e-1}}$ thorough algorithm update replay buffer $\mathcal{D}$.
4:     Sample a minibatch of $(\boldsymbol{s}, \boldsymbol{a}, r, \boldsymbol{s}')$ from $\mathcal{D}$.
5:     *# Policy evaluation.*
6:     Sample $\boldsymbol{a}'$ via reverse diffusion process (8) with $s_{\theta_{e-1}}$.
7:     Update $Q_e$ with soft policy evaluation (4).
8:     *# Policy improvement for diffusion policies.*
9:     Sample $t$ uniformly from $\{1, 2, \ldots, T\}$. Sample $\boldsymbol{a}_t$ from $h_t$.
10:    Sample $K$ $\tilde{\boldsymbol{a}}_0^{(i)} \sim \phi_{0|t}$ for $i = 1, 2, \ldots, K$.
11:    Compute $Q_e(\boldsymbol{s}, \tilde{\boldsymbol{a}}_0)$ and update $\theta_e$ with empirical loss of

$$\mathbb{E}_{\boldsymbol{s},t} \left[ \frac{1}{K} \exp\left( Q_e\left(s, \tilde{\boldsymbol{a}}_0^{(i)}/\lambda_e\right) - \log \sum_i \exp\left( Q_e\left(s, \tilde{\boldsymbol{a}}_0^{(i)}\right)/\lambda_e\right)\right) \left\| s_\theta\left(\boldsymbol{a}_t; \boldsymbol{s}, t\right) - \nabla_{\boldsymbol{a}_t} \log \phi_t\left(\tilde{\boldsymbol{a}}_0 \,|\, \boldsymbol{a}_t\right) \right\|^2 \right]$$

.

12:    Update KL-divergence coefficient $\lambda_e \leftarrow \lambda_{e-1} + \beta(\lambda_{e-1} - \lambda_{\text{target}})$.
13: **end for**

---

where the RSSM optimizes

$$\mathbb{E}_{\substack{\mathbf{x}_t \sim \tilde{p} \\ \mathbf{x}_0 \sim \phi_{0|t}}} \left[ p_0(\tilde{\mathbf{x}}_0) \left\| s_\theta\left(\mathbf{x}_t; t\right) - \nabla_{\mathbf{x}_t} \log \phi_{0|t}(\tilde{\mathbf{x}}_0 | \mathbf{x}_t) \right\|^2 \right] \tag{31}$$

for the Gaussian sampling, $\tilde{p}(\mathbf{x}_t) = \mathcal{N}(0, 4\mathbf{I})$ for all $t$ and for uniform sampling, $\tilde{p}_t$ is a uniform distribution from $[-6, 6]$ on both dimensions. The score network is trained via the hyperparameters listed in Table 3. The Langevin dynamics has direct access to the true score function $\nabla_{\mathbf{x}} \log p_0(\mathbf{x})$.

*Table 3.* Hyperparameters for the toy example.

| Name | Value | Name | Value |
|------|-------|------|-------|
| Learning rate | 3e-4 | Diffusion noise schedule | linear |
| Diffusion steps | 20 | Diffusion noise schedule start | 0.001 |
| Hidden layers | 2 | Diffusion noise schedule end | 0.999 |
| Hidden layer neurons | 128 | Training batch size | 1024 |
| Activation Function | LeakyReLU | Training epoches | 300 |

### C.4. Additional Toy Examples

Additionally, we show a more complex two-moon distribution with a known energy function as

$$\log p(z) = -\frac{1}{2}\left(\frac{\|z\| - 2}{0.2}\right)^2 + \log\left(\exp\left(-\frac{1}{2}\left(\frac{z_1 - 2}{0.3}\right)^2\right) + \exp\left(-\frac{1}{2}\left(\frac{z_1 + 2}{0.3}\right)^2\right)\right) \tag{32}$$

We compare the proposed `RSM` with DDPM (Ho et al., 2020) that has access to the data samples, and two other Boltzmann samplers that have access to the energy functions (32), iDEM (Akhound-Sadegh et al., 2024) based on diffusion with approximated noise-perturbed score functions, and FAB (Midgley et al., 2022) based on normalizing flows. The results are shown in Figure 5. We can see that three diffusion-based methods, RSM, DDPM, and iDEM, show similar performance. However, the two set of data samples are not separate enough for FAB, showing the advantage of the diffusion model and the limitations of normalizing flow from the restrictive invertible mappings.

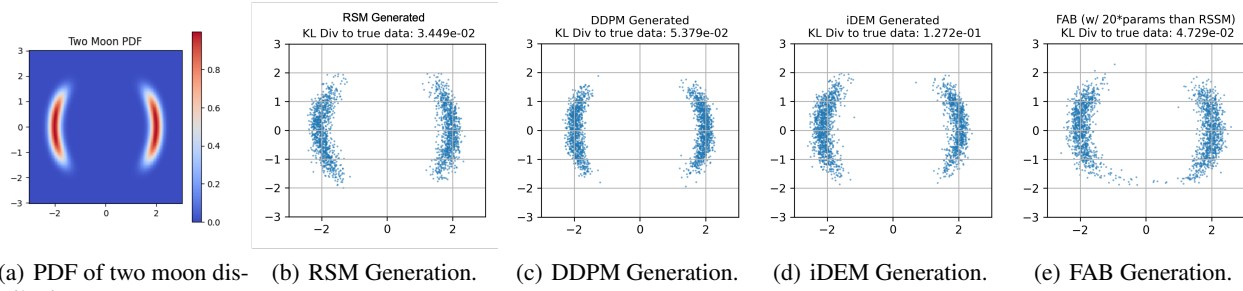

(a) PDF of two moon distribution.    (b) RSM Generation.    (c) DDPM Generation.    (d) iDEM Generation.    (e) FAB Generation.

*Figure 5.* Results to Fit Two Moon distribution, a commonly used Boltzmann sampler benchmark. We compare RSM, DDPM, iDEM, and FAB. RSM, DDPM, and iDEM all recover two separate modes, while FAB shows connections between the two modes.

### C.5. Baselines

We include two families of methods as our baselines. For the first family of methods, we select 5 online diffusion-policy RL algorithms: QSM (Psenka et al., 2023), QVPO (Ding et al., 2024a), DACER (Wang et al., 2024), DIPO (Yang et al., 2023) and DPPO (Ren et al., 2024). We include both off-policy (QSM, QVPO, DACER, DIPO) and on-policy (DPPO) diffusion RL methods among this group of algorithms. QSM uses the Langevin dynamics, one of the MCMC methods, with derivatives of learned $Q$-function as the score function. QVPO derives a Q-weighted variational objective for diffusion policy training, yet this objective cannot handle negative rewards properly. DACER directly backward the gradient through the reverse diffusion process and proposes a GMM entropy regulator to balance exploration and exploitation. DIPO utilizes a two-stage strategy, which maintains a large number of state-action particles updated by the gradient of the Q-function, and then fit the particles with a diffusion model. DPPO constructs a two-layer MDP with diffusion steps and environment steps, respectively, and then performs Proximal Policy Optimization on the overall MDP. In our experiments, we use the training-from-scratch setting of DPPO to ensure consistency with other methods.

The second family of baselines includes 3 classic model-free RL methods: PPO (Schulman et al., 2017), TD3 (Fujimoto et al., 2018) and SAC (Haarnoja et al., 2018). For PPO, we set the replay buffer size as 4096 and use every collected sample 10 times for gradient update. Across all baselines, we collect samples from 5 parallel environments in a total of 1 million environment interactions and 200k epoches/iterations. The results are evaluated with the average return of 20 episodes across 5 random seeds.

### C.6. Hyperparameters

*Table 4.* Hyperparameters

| Name | Value |
| --- | :---: |
| Critic learning rate | 3e-4 |
| Policy learning rate | 3e-4, linear annealing to 3e-5 |
| Diffusion steps | 20 |
| Diffusion noise schedules | Cosine |
| Policy network hidden layers | 3 |
| Policy network hidden neurons | 256 |
| Policy network activation | Mish |
| Value network hidden layers | 3 |
| Value network hidden neurons | 256 |
| Value network activation | Mish |
| Replay buffer size (off-policy only) | 1 million |

where the Cosine noise schedule means $\beta_t = 1 - \frac{\bar{\alpha}_t}{\alpha_{t-1}}$ with $\bar{\alpha}_t = \frac{f(t)}{f(0)}$, $f(t) = \cos\left(\frac{t/T+s}{1+s} * \frac{\pi}{2}\right)^2$.

