# OpenReview forum: "Efficient Online Reinforcement Learning for Diffusion Policy"
_ICML.cc/2025/Conference — ICML 2025 poster_

### Official Review · Reviewer_bxjo · 2025-03-10

**Overall Recommendation:** 3

**Summary:**

This paper studies training diffusion policy in the online RL setting and proposes an algorithm based on the energy-based view of diffusion models. It then evaluates the proposed algorithm on Mujoco tasks provided in Gym.

# Main Ideas
* To learn a diffusion policy, we need to learn the score function $s_\theta(a_t;s,t)$.
* This paper leverage the energy-based view of diffusion models: the score function $s_\theta(a_t;s,t)$ matches the noise-perturbed score function $\nabla_{a_t}\log\tilde{\pi}_t(a_t|s)$  in expectation when considering an energy-based model. So the score function can be learned with samples $a_0$ and $a_t$.

* The noise-perturbed score function can be expressed as the expectation of $ q_{t|0} (a_t | a_0) $  with regard to the distribution of the energy based model.

* Given samples $a_t$, we can approximate such expectation with the reverse sampling distribution $\tilde{q}_{0|t}$ and the Q-function.

# Main Results

* The authors show that their proposed reverse sampling score matching can be used to fit a diffusion model on a 2D Gaussian dataset.
* Their proposed policy learning algorithm SDAC outperforms non-diffusion RL methods and recent methods for training diffusion policy in online RL.
* The SDAC is more robust than other diffusion-based methods in the sense that it achieves consistent performance on 10 tasks.

**Claims And Evidence:**

The main claims are reverse sampling score matching (RSSM) and a practical algorithm SDAC.
Among the explanation of RSSM, I found the reverse sampling trick (14) introduced in line 213 hard to understand and seems lacking any evidence or explanations.

**Essential References Not Discussed:**

No.

**Experimental Designs Or Analyses:**

Yes, I have checked the results in section 5.1 and 5.2. In section 5.2, the authors do not provide results for DPPO, TD3 or PPO. I think it is better to provide results for all the baselines, at least for TD3, because TD3 is reported to be consistently outperforming SAC on Mujoco Tasks.

**Methods And Evaluation Criteria:**

The authors use ten continuous control tasks provided by the GYM repository, which is standard for online RL problems.

**Other Comments Or Suggestions:**

No.

**Other Strengths And Weaknesses:**

My remaining comments are mostly about the clarity of the paper.

1. The paper lacks an intuitive explanation for why the proposed RSSM can work. Figure 1 is actually not very informative, as its messages are merely (i) the Q-function is approximated with policy evaluation (ii) the loss of the actor involves the approximated Q-function. I would suggest the authors to provide a figure that explains the main ideas summarized above and emphasize that the loss of the actor can be computed with only diffusion samples.

2. The sampling distribution $\tilde{p}_t$ in Theorem 3.2 is introduced without any explanation, which is very confusion. It is better to, at least explain that, it can be approximated with diffusion samples.

3. The related work of this paper is scattered throughout the paper, which breaks the flow of this paper.

4. It is better to put the "Difficulties to train diffusion model in online RL setup." before 3.1, as they are not related to the energy-based view of diffusion models but with the main challenge to be resolved in this paper.

**Questions For Authors:**

1. Could you please elaborate on the reverse sampling trick?
2. Could you please provide results for all the baselines mentioned in 5.2.1?
3. Could you please analyze why TD3 cannot outperform SAC?

**Relation To Broader Scientific Literature:**

While the authors argue that the proposed RSSM can applied to *any probabilistic model problems with known energy function*, the Gaussian mixture task is way to simple for support this. In this sense, it seems that the proposed methods are quite restricted to RL problems that learn a soft Q function.

**Theoretical Claims:**

No, I have not check the correctness of any proofs. The proof scratch in page 4 is hard to follow due to the missing explanation for the reverse sampling trick.

---

> ### Author Rebuttal · Authors · 2025-04-01
>
> We appreciate the reviewer’s thoughtful comments. Here are the responses, where we grouped relevant questions together,
>
> ## Q1: Explanation about the reverse sampling trick.
>
> > *I found the reverse sampling trick (14) introduced in line 213 hard to understand and seems lacking any evidence or explanations.*
>
> We provide the detailed derivations of the reverse sampling trick in our response to Reviewer pogC.
>
> The proposed RSSM loss and original DDPM loss [1] both aim to train a diffusion model to generate data from the target distribution $p_0$. However, DDPM loss requires sampling from $p_0$, which is not possible in online RL since we only know the energy function of $p_0$.
>
> The core novelty of RSSM is a tractable approach to train diffusion model/policy purely from the energy function rather than samples. The reverse sampling trick is an algebric operation to use sampling-based approximation to calculate the loss.
>
>
>
> > *The sampling distribution $\tilde{p}_t$ in Theorem 3.2 is introduced without any explanation, which is very confusion. It is better to, at least explain that, it can be approximated with diffusion samples.*
>
> The sampling distribution $\tilde{p}_t$ is a distribution we can choose. Any distributions covering the support $p_t(x_t)$ can be used, such as Gaussian, Cauchy, etc.
>
> Being approximated by diffusion models is not required for $\tilde{p}_t$.
>
> **The choice of $\tilde{p}_t$**. Our choice of $\tilde{p}_t$ is based on the fact that the RSSM loss relies on the energy function ($Q$-function in RL) that is more accurate as samples of $a$ are close to the current policy. Therefore, we directly use the current reverse sampling at the $t$ step, which is exactly the current policy.
>
> [1] Ho, J., Jain, A., & Abbeel, P. Denoising diffusion probabilistic models. NeurIPS 2020.
>
>
> ## Q2:
>
> > *In section 5.2, the authors do not provide results for DPPO, TD3 or PPO. I think it is better to provide results for all the baselines...*
>
> We report the results for all the baselines, the reviewer might mean we did not provide the training curves in Figure 2 for DPPO, TD3, and PPO. We reported in [Figure 8 here](https://anonymous.4open.science/r/rebuttal_pdf-D49E/rebuttal_figures.pdf).
>
>
> ## Q3 Performance comparison between TD3 and SAC
>
> > *because TD3 is reported to be consistently outperforming SAC on Mujoco Tasks.*
>
> > *Could you please analyze why TD3 cannot outperform SAC?*
>
>
> In Table 1,
> - TD3 is better in Reacher, Humanoid, Pusher, Ant, and Swimmer.
> - SAC is better in HalfCheetah, Hopper, and Walker2d,
> - They are very close in InvertedDoublePendulum and InvertedPendulum.
>
> Therefore, it is difficult to say one algorithm outperforms another one.
>
> Generally speaking, the performance gap between TD3 and SAC is sensitive to various hyperparameters. We reviewed some of our references using both as baselines and summarized the comparison in the figure below.
>
>
> | Reference | Tasks                  | TD3 is better  | SAC is better | Total |
> |-----------|------------------------|----------------|---------------|-------|
> | QVPO   | Gym MuJoCo             | 2              | 3             | 5     |
> | QSM    | Deepmind Control Suite | 4              | 4             | 8     |
> | DACER  | Gym MuJoCo             | 1              | 7             | 8     |
> | DIPO   | Gym MuJoCo             | 2              | 3             | 5     |
>
> Therefore, it is generally not straightforward to conclude that SAC consistently outperforms TD3, or vice versa.
>
>
>
> ## Q4: about the toy example
>
> > *The author argue that RSSM can apply to any probablistic model with known enery function. The Gaussian mixture task is way to simple for support this... it seems that the proposed methods are quite restricted to RL problems...*
>
> **A4:** Our toy example on Gaussian mixture is just a proof-of-concept, not for showing the capability limit of RSSM.
> Moreover, modeling a Gaussian mixture is non-trivial since [1] highlights its slow-mixing issue, and our Figure 2(e) further shows that naive Langevin dynamics fails to recover the correct mixture within finite steps.
>
> To further demonstrate RSSM’s effectiveness, we include results on the Two Moon distribution, a standard benchmark for Boltzmann samplers, in [Figure 6](https://anonymous.4open.science/r/rebuttal_pdf-D49E/rebuttal_figures.pdf). We also add iDEM [2] and FAB [3] as baselines. RSSM performs well and achieves the lowest KL divergence among all methods.
>
> [1] Song, Yang, and Stefano Ermon. Generative modeling by estimating gradients of the data distribution.
>
> [2] Akhound-Sadegh, Tara, et al. Iterated denoising energy matching for sampling from boltzmann densities.
>
> [3] Midgley, Laurence Illing, et al. Flow Annealed Importance Sampling Bootstrap.
>
>
> ---
>
> We appreciate your valuable comments and hope our responses have clarified the concerns raised. We would be grateful if you would consider updating your score in light of these clarifications.

---

### Official Review · Reviewer_e72f · 2025-03-15

**Overall Recommendation:** 3

**Summary:**

This work presents a diffusion-based online RL method called Soft Diffusion Actor-Critic (SDAC). The authors highlight the difficulty of training online RL methods due to inability to sample from the target distribution (optimal policy) and the computationally intensive nature of training some diffusion-based approaches. The authors utilize the connection between diffusion models and EBMs to motivate their approach of sampling from energy-based policies. Their approach, reverse sampling score matching, is theoretically motivated by deriving the loss function and showing it learns the correct score function. Experiments are performed on a simple 2-D Gaussian mixture to validate their diffusion-based approach, then online RL experiments are performed on the Mujoco benchmark which demonstrates the improved performance of SDAC compared to existing classical and diffusion-based model-free methods.

## Update after rebuttal
The authors' response addressed most of my major concerns, hence I raised my score. The explanation for RSSM in the original paper was incorrect/misleading, hence the authors should incorporate their latest response below in the update manuscript.

**Claims And Evidence:**

- The claim that SDAC is “more efficient compared to recent diffusion policies for online RL” (line 315-317) is not supported by enough theoretical or empirical evidence. The authors claim that their method avoids sampling from $\pi_\text{target}$ and the training loss has similar cost as the denoising score-matching loss, but both of these statements apply for [1] (which uses iDEM [2] and has been mischaracterized by the authors in line 321-325) as well as [3]. The conclusions drawn from the experiment in Table 2 which compares memory and wall clock time of different methods also have several issues, see ‘Experimental Designs Or Analyses’ below.
- The claim that “performance is increased by more than 120% over SAC” is slightly misleading and there are certain caveats that must be clarified. First, based on the results in Section 5.2, this statement seems to be true for only 2 out of the 10 environments considered in the experiments. Second, the standard for off-policy methods is to perform one network update per environment step, however, in the experiments all methods perform 200k iterations for 1 million environment steps. Based on public benchmarks of SAC on Mujoco (https://spinningup.openai.com/en/latest/spinningup/bench.html), it seems after 1 million updates, SAC does reach performance levels on par or higher than those claimed for SDAC.

[1] Jain, Vineet, Tara Akhound-Sadegh, and Siamak Ravanbakhsh. "Sampling from Energy-based Policies using Diffusion." *arXiv preprint arXiv:2410.01312* (2024).

[2] Akhound-Sadegh, Tara, Jarrid Rector-Brooks, Avishek Joey Bose, Sarthak Mittal, Pablo Lemos, Cheng-Hao Liu, Marcin Sendera et al. "Iterated denoising energy matching for sampling from boltzmann densities." In *Proceedings of the 41st International Conference on Machine Learning*, pp. 760-786. 2024.

[3] Yang, Long, Zhixiong Huang, Fenghao Lei, Yucun Zhong, Yiming Yang, Cong Fang, Shiting Wen, Binbin Zhou, and Zhouchen Lin. "Policy representation via diffusion probability model for reinforcement learning." *arXiv preprint arXiv:2305.13122* (2023).

**Essential References Not Discussed:**

The paper cites most relevant works in the area of diffusion models applied for online RL. However, the descriptions of some of these works are completely incorrect, and the authors fail to acknowledge that some of the issues they tackle in this work have also been addressed by existing works.

- The desription of [1] is completely incorrect, in that it does not use Langevin sampling, rather it uses a diffusion process (specifically iDEM) to learn a policy that samples from the Boltzmann distribution of the Q-function.
- The claim of the authors that [2] “induce huge memory and computation costs” is not elaborated upon sufficiently. The approach updates the actions stored in the replay buffer (NOT as a set of particles as claimed by the authors) based on the gradient of the Q-function, and fits a diffusion model using score matching loss to these updated actions. The memory footprint should not be significantly different from the proposed method.
- The authors mention that the necessity of sampling from target distribution and backpropagating through the diffusion chain are two main issues with employing diffusion-based methods. However, the authors fail to acknowledge that several existing approaches [1,2] have been proposed that tackle these issues.

[1] Jain, Vineet, Tara Akhound-Sadegh, and Siamak Ravanbakhsh. "Sampling from Energy-based Policies using Diffusion." *arXiv preprint arXiv:2410.01312* (2024).

[2] Yang, Long, Zhixiong Huang, Fenghao Lei, Yucun Zhong, Yiming Yang, Cong Fang, Shiting Wen, Binbin Zhou, and Zhouchen Lin. "Policy representation via diffusion probability model for reinforcement learning." *arXiv preprint arXiv:2305.13122* (2023).

**Experimental Designs Or Analyses:**

- The overall experimental design seems to be sound. There is the question of how would SDAC perform in relation to baselines if the standard practice of 1 million updates for 1 million environment steps were used.
- The conclusions drawn from the benchmarking of memory and wall clock time seem misleading. In any implementation, there are several factors that can influence the memory and time, including the deep learning framework used, third party libraries, how optimized the implementations are etc. I believe it is incorrect to directly draw the conclusion that some method is more efficient without accounting for these factors. The authors should provide specific reasons why other baseline methods have higher memory/time that SDAC.

**Methods And Evaluation Criteria:**

The authors consider the Mujoco benchmark as their main experimental setting, which is a popular suite of environments used for comparing RL algorithms. The baselines used seem to be exhaustive. There are two minor improvements I can suggest:

- Mujoco benchmark has been considered saturated for some time, so it would be nice if the authors could demonstrate performance improvements on more settings to strengthen their claims.
- The 2-D Gaussian experiments were meant as a proof-of-concept, but it would be beneficial to include other Boltzmann samplers [1,2,3] as baselines.

[1] Akhound-Sadegh, Tara, Jarrid Rector-Brooks, Avishek Joey Bose, Sarthak Mittal, Pablo Lemos, Cheng-Hao Liu, Marcin Sendera et al. "Iterated denoising energy matching for sampling from boltzmann densities." In *Proceedings of the 41st International Conference on Machine Learning*, pp. 760-786. 2024.

[2] Midgley, Laurence Illing, Vincent Stimper, Gregor NC Simm, Bernhard Schölkopf, and José Miguel Hernández-Lobato. "Flow Annealed Importance Sampling Bootstrap." In *The Eleventh International Conference on Learning Representations*.

[3] He, Jiajun, Wenlin Chen, Mingtian Zhang, David Barber, and José Miguel Hernández-Lobato. "Training Neural Samplers with Reverse Diffusive KL Divergence." In *The 28th International Conference on Artificial Intelligence and Statistics*.

**Other Comments Or Suggestions:**

- The writing at the sentence level, particularly in the introduction could be greatly improved. Some examples of poor writing,
    - Line 42-43: Huge successes of diffusion-based generative models have been witnessed recently
    - Line 46-47: diffusion models achieved superior expressiveness and multimodality
    - Line 13: naturally benefit ~~the~~ policies
    - Line 16: offline RL, where expert datasets are presented
    - There are many more examples, I suggest the authors revise the writing of the paper carefully
- Missing one citation reference in Appendix A.

**Other Strengths And Weaknesses:**

- The proposed method claims to use the maximum entropy RL formulation via soft policy evaluation, but it is not clear how $\log \pi(a_t | s_t)$ is calculated efficiently in equation (2), since calculating  exact log likelihoods for diffusion models is a non-trivial (and computationally demanding) problem.
- The overall structuring and presentation of the paper could be improved. Section 1 and 2 do a good job of introducing the main problem and describing the setting. From Section 3 onwards, it is easy to get lost in the mathematical details of the proposed method. It would be easier for the reader if the authors first present their method clearly describing the sampling and training procedures, then proceed to describe the theoretical results.

**Questions For Authors:**

- In Section 4.1, why is there a need to add Gaussian noise for exploration? In principle, using a temperature parameter to scale the energy function should be enough since high temperatures would correspond to more random behavior. This is also the approach used in other energy-based methods like [1,2].
- What is the benefit of the proposed RSSM method compared to other Boltzmann sampling methods such as iDEM [3] and FAB [4]?
- How does SDAC compare to other methods when the number of iterations are increased to 1M steps (which is commonly used in online RL experiments)?

[1] Haarnoja, Tuomas, Aurick Zhou, Pieter Abbeel, and Sergey Levine. "Soft actor-critic: Off-policy maximum entropy deep reinforcement learning with a stochastic actor." In *International conference on machine learning*, pp. 1861-1870. Pmlr, 2018.

[2] Jain, Vineet, Tara Akhound-Sadegh, and Siamak Ravanbakhsh. "Sampling from Energy-based Policies using Diffusion." *arXiv preprint arXiv:2410.01312* (2024).

[3] Akhound-Sadegh, Tara, Jarrid Rector-Brooks, Avishek Joey Bose, Sarthak Mittal, Pablo Lemos, Cheng-Hao Liu, Marcin Sendera et al. "Iterated denoising energy matching for sampling from boltzmann densities." In *Proceedings of the 41st International Conference on Machine Learning*, pp. 760-786. 2024.

[4] Midgley, Laurence Illing, Vincent Stimper, Gregor NC Simm, Bernhard Schölkopf, and José Miguel Hernández-Lobato. "Flow Annealed Importance Sampling Bootstrap." In *The Eleventh International Conference on Learning Representations*.

**Relation To Broader Scientific Literature:**

The paper is one of many which aims to tackle challenges in online RL, specifically related to learning more expressive policy functions, by using a diffusion model. This paper attempts to solve this problem by proposing a novel loss function that bypasses the issue of sampling from a target distribution, and aims to reduce computation during the backward pass.

While the specific ideas introduced in this work are novel, there are other existing works which tackle the same problems in diffusion-based policies for online RL [1,2].

[1] Yang, Long, Zhixiong Huang, Fenghao Lei, Yucun Zhong, Yiming Yang, Cong Fang, Shiting Wen, Binbin Zhou, and Zhouchen Lin. "Policy representation via diffusion probability model for reinforcement learning." *arXiv preprint arXiv:2305.13122* (2023).

[2] Jain, Vineet, Tara Akhound-Sadegh, and Siamak Ravanbakhsh. "Sampling from Energy-based Policies using Diffusion." *arXiv preprint arXiv:2410.01312* (2024).

**Theoretical Claims:**

I checked the proofs provided in the main paper as well as the appendix. One of the key ideas presented in the paper, the reverse sampling trick which is also used in Theorem 3.2, incorrectly replaces the distribution $q_{t | 0} (\cdot | a_0)$ with $\tilde{q}_{0|t}(\cdot | a_t)$ in the loss function based on algebraic manipulation of the Gaussian pdf. However, from Bayes’ rule,

$$
q_{t|0}(a_t | a_0) = \frac{q_{0|t}(a_0 | a_t) q(a_t)}{q(a_0)},
$$

where $q(a_t)$ and $q(a_0)$ are marginal distributions for $a_t$ and $a_0$ respectively. Since equation (13) is integrated wrt both $a_t$ and $a_0$, we cannot simply ignore these terms. In addition, when comparing their score functions, the term $q(a_t)$ cannot simply be ignored and hence the score function $\nabla_{a_t} \log q_{t|0} (a_t | a_0)$ cannot be replaced with  $\nabla_{a_t} \log \tilde{q}_{0|t} (a_0 | a_t)$.

Overall, it seems the authors fail to realize that the pdf $q_{t | 0} (\cdot | a_0)$ is not just an algebraic expression but has a semantic meaning associated with it, where $a_0$ is the mean and $a_t$ is the random variable, which is NOT the same as $\tilde{q}_{0|t}(\cdot | a_t)$ where $a_t$ is the fixed mean and $a_0$ is the random variable.

---

> ### Author Rebuttal · Authors · 2025-04-01
>
> We thank the reviewer for the constructive feedback. Below, we provide our response and have merged some questions and reindexed the references for clarity.
>
>
> ## Claims And Evidence
>
> **Q1:**
> > *reasons why other baseline methods have higher memory/time than SDAC.*
>
> > *benefit compared to other Boltzmann sampling methods such as iDEM [3] and FAB [4]?*
>
>   1. DQS/iDEM [1,2] use $K$ Monte-Carlo samples and $K$ energy function evaluations for one $\nabla\log(x_t)$. RSSM only needs one sample and energy function evaluation, saving memory and time.
>   2. The sampling in RSSM is unbiased, while iDEM sampling is biased according to Proposition 1 in [2].
>   3. DQS/iDEM/QSM use the gradient of energy function, which might be inaccurate when the energy function (Q-function) is learned in RL.
>   4. FAB [4] leverages flow models whose expressiveness is limited compared to diffusion-based models. FAB also suffers from high variance due to importance sampling.
>
> > *Add Boltzmann samplers as toy example baseline.*
>
>   We add iDEM and FAB to the toy example, as well as the two-moon distribution in [Figure 5, 6](https://anonymous.4open.science/r/rebuttal_pdf-D49E/rebuttal_figures.pdf). We also count the training time and memory footprint in Table 5.
>
> > *The claim [3] “induce huge memory and computation costs” is not elaborated upon sufficiently. [3] updates the actions stored in the replay buffer (NOT as a set of particles as claimed by the authors)...*
>
> We respectfully disagree with the reviewer’s claim. According to the official DIPO implementation [3], `line 116` in `main.py` explicitly calls `diffusion_memory = DiffusionMemory(state_size, action_size, memory_size, device)`, clearly indicating the use of an additional buffer, inducing memory and time cost.
>
> - QVPO,DACER,DIPO: discussed in Section 4.2.
>
> ---
>
> **Q2:** Performance not comparable to spinning up SAC at 1 million updates.
>
> **A2:**  Spinning Up’s results run for **3 million** updates, not 1M as claimed. At 1M, it often underperforms SDAC, which uses only 200K updates.
>
> ---
>
> ## Methods And Evaluation Criteria
> **Q3:** Experiments on other tasks.
>
> - Please refer to [Figure 4 & Table 4](https://anonymous.4open.science/r/rebuttal_pdf-D49E/rebuttal_figures.pdf). We select three tasks from DeepMind Control Suite, and a PushT task that is common in diffusion policy.
>
> ---
>
> ## Theoritical claims
>
> **Q4:** RSSM does not satisfy Bayes' rule.
>
> **A4:** We clarify that the reverse sampling distribution **$\tilde q_{0|t}$ is not, and does not intend to approximate the posterior distribution $q_{0|t}$ in the reverse process mentioned by the reviewer.** $ q_{0|t}$ is usually intractable, while $\tilde q_{0|t}(\cdot|a_t)$ is a Gaussian.
>
> We respectfully argue that we did not ignore any terms. In fact, our derivation does not use the Bayes' rule, it is based on the probability density function of Gaussians. **We provide the detailed derivations in our response to Reviewer pogC**, including the score function equivalence.
>
> The novelty of RSSM is integrating score matching loss (12) over measure $g$ in (31), so that it is tractable through unbiased sampling-based approximation, when we cannot sample from $p_0$. We realize that the current notations easily cause causality confusion. We will carefully revise the notations in the revision.
>
> ---
>
> ## Experimental Designs
>
> Please see [Figure 1 and Table 1](https://anonymous.4open.science/r/rebuttal_pdf-D49E/rebuttal_figures.pdf) for performance of 1M updates.
>
> ## Essential References Not Discussed
>
> **Q5:**
> > *The description of [1] is completely incorrect, in that it does not use Langevin sampling, rather it uses a diffusion process...*
>
> > *the authors fail to acknowledge that several existing approaches [1,2] that tackle these issues.*
>
> We acknowledge a partial inaccuracy in our claim about [1]. Nonetheless, one of the core issues we highlight remains: the gradient of learned Q-functions might be inaccurate, affecting RL performance.
>
> We did miss the literature on Boltzmann samplers. We will address this in the revision.
>
> ---
>
> ## Other questions:
>
> **Q6** Computing the log probability, the necessity of additive Gaussian noise, and using temperature parameters.
>
> **A6** Please refer to **A3,A4** in rebuttal to reviewer pogC. We use temperature parameters to control the additive noise scale.
>
> Thanks for pointing out the typos and writing comments. We will revise accordingly.
>
> [1] Jain, Vineet, et al. Sampling from Energy-based Policies using Diffusion.
>
> [2] Akhound-Sadegh, Tara, et al. Iterated denoising energy matching for sampling from boltzmann densities.
>
> [3] Midgley, Laurence Illing, et al. Flow Annealed Importance Sampling Bootstrap.
>
> [4] Yang, Long, et, al. Policy representation via diffusion probability model for reinforcement learning.
>
> ---
>
> We appreciate your valuable comments and hope our responses have clarified the concerns raised. We would be grateful if you would consider updating your score in light of these clarifications.

---

> > ### Comment · Reviewer_e72f · 2025-04-03
> >
> > I thank the reviewers for their response as well as the new experiments. I appreciate that the authors added a comparison with Boltzmann samplers, experiments with more environments, and running methods with 1M updates. I also acknowledge their responses to my concerns with the efficiency of RSSM, benefits compared to other methods and their responses to my questions about log probability calculation and temperature.
> >
> > > We respectfully disagree with the reviewer’s claim. According to the official DIPO implementation [3], line 116 in `main.py` explicitly calls `diffusion_memory = DiffusionMemory(state_size, action_size, memory_size, device)`, clearly indicating the use of an additional buffer, inducing memory and time cost.
> >
> > In my review I had stated: "The claim DIPO [3] 'induce huge memory and computation costs' is not elaborated upon sufficiently. The approach updates the actions stored in the replay buffer (NOT as a set of particles as claimed by the authors)". The authors seem not to have read my comment carefully, **I do not challenge that DIPO is not memory intensive; I merely ask them to correct the inaccuracy** since maintaining a set of particles (used in the context of sampling to denote a current population of samples) is different from a replay buffer (used to store previously generated samples to train a model).
> >
> > > A2: Spinning Up’s results run for 3 million updates, not 1M as claimed. At 1M, it often underperforms SDAC, which uses only 200K updates.
> >
> > The authors misread my comment. I am well aware that spinning up runs their experiments for 3M updates. As stated in my review “it seems after 1 million updates”, one can surmise the performance of SAC at 1M steps from the training runs. And “At 1M, it often underperforms SDAC, which uses only 200K updates” is simply not true. Looking from the spinning up documentation the scores for SAC at 1M steps: HalfCheetah ~11000, Hopper ~3500, Walker2d ~4000, Ant ~4000. Only Ant is significantly worse than SDAC, the rest are comparable to the results in the paper, so the author's statement is simply untrue. Nevertheless, the new experiments with 1M updates are appreciated and they seem to align with spinning up.
> >
> > My main point in that statement was that the language of the paper overclaims the performance gains, since the gain of 120% over SAC is only valid for 2 out of 10 experiments from the results in the paper and after the new results with 1M updates the relative performance gains are reduced. No one is disputing that SDAC does seem to perform well compared to baselines, but **the authors misrepresent their performance - a concern they seem to have ignored**.
> >
> > > We respectfully argue that we did not ignore any terms. In fact, our derivation does not use the Bayes' rule, it is based on the probability density function of Gaussians. We provide the detailed derivations in our response to Reviewer pogC, including the score function equivalence.
> >
> > **I understand and clearly state in my review that the authors have used algebraic manipulation of the Gaussian pdf to obtain this expression and they do not use Bayes' rule, which is exactly the problem**. There is a reason Bayes' rule exists, and why diffusion-based Boltzmann samplers are tricky to train because of the need to sample from the conditional distribution $q_{t|0}(a_t | a_0)$.
> >
> > As I mentioned in my review, there is a semantic meaning associated with the expressions. $q_{t|0}(a_t | a_0)$ involves sampling $a_t$ based on some known $a_0$, whereas $\tilde{q}_{0|t}(a_0 | a_t)$ involves sampling $a_0$ based on some known $a_t$. These two things are **not** the same and cannot be substituted. The correct way to substitute this sampling would be to use Bayes' rule to approximate the posterior, or to use importance sampling techniques (which would require calculating appropriate weights).
> >
> > ---
> >
> > I maintain my current rating due to major concerns being unaddressed. If the authors can respond to my concerns satisfactorily, I am willing to update my score. I repeat the main points from my review:
> >
> > - The authors overstated claims related to the performance of their method and misquote the performance of baselines.
> > - The theoretical justification for their method seems to be incorrect. My argument remains the same as in my review but the authors seem to have misunderstood.
> > - Relatively minor points: correctly referring to some baselines and issues with accuracy of measuring wall clock times.
> >
> > ------
> >
> > Edit: The authors' reply to this rebuttal clarifies the main issue that was the derivation and the reasoning behind RSSM. The version they describe below is sound and this should be clearly explained in the paper to avoid misunderstanding. I have no major concerns remaining, hence I update my score.

---

> > > ### Author Response · Authors · 2025-04-07
> > >
> > > We thank the reviewer for the detailed reply and apologize for the confusion. Due to the space limit, we cannot include many details in the previous rebuttal. We add the following clarifications.
> > >
> > > **Notation**: Given $t$, $s$, the joint distribution $p(a_0, a_t|s)$ is,
> > > $$
> > > p(a_0, a_t|s) = \pi(a_0|s)q_{t|0}(a_t|a_0) = p_t(a_t|s)q_{0|t}(a_0|a_t)
> > > $$
> > > where the marginals are policy $\pi(a_0|s)\propto \exp(Q(s, a_0)/\lambda)$ and the perturbed policy at $t$-step, $p_t(a_t|s) = \int \pi(a_0|s)q_{t|0}(a_t|a_0)da_0$.
> > >
> > > ## 1.  The semantic meanings and Bayes' rule---RSSM as weighted DSM loss
> > >
> > > We realize the `replace` easily causes confusion. We first explain our core idea of RSSM, and discuss the confusion about semantic meanings and where the `replace` happens.
> > >
> > >  ### Core ideas:
> > >
> > > **The RSSM did not try to sample from the joint distribution $p(a_0, a_t|s)$. RSSM loss function is designed as an integral on a different measure $h(a_0,a_t|s) =\exp(Q(s, a_0) / \lambda) \tilde p_t(a_t|s)  \tilde q_{0|t}(a_0 | a_t)$, while maintaining the same optimal solution $s_\theta(a_t;t,s) = \nabla_{a_t}\log p_t(a_t|s)$.**
> > >
> > > The new measure $h$ admits our reverse sampling rules weighted by factor $\exp(Q(s, a_0)/\lambda)$.
> > >
> > > We explain the details starting from the commonly used denoising score matching (DSM) loss,
> > >
> > > ### Original DSM loss and optimal solution
> > >
> > > Given $t, s$, the DSM loss in Eq. (5) is
> > > $$
> > > L_{\rm DSM}(\theta; t, s) = \iint ||s_\theta(a_t; t, s)-\nabla_{a_t}\log q_{t|0}(a_t|a_0)||^2 p(a_0,a_t | s)da_0da_t \quad (*)
> > > $$
> > > According to Appendix Proposition 1 (line 606-617),  the optimal solution is achieved when $s_\theta$ matches the score function,
> > > $$
> > > s_\theta(a_t; t, s)=\nabla_{a_t}\log p_t(a_t|s),\forall a_t
> > > $$
> > >
> > > However, the DSM loss in intractable since
> > > 1. We cannot sample from $\pi$, so we cannot sample first $a_0\sim\pi$ then $a_t\sim q_{t|0}$.
> > > 2. posterior $q_{0|t}$ is unknown, so we cannot first sample first $a_t\sim p_t$ then $a_t\sim q_{0|t}$.
> > >
> > >
> > > ### RSSM as weighted DSM loss
> > >
> > > To avoid the intractability, we select a custom measure $h(a_0,a_t|s)$ to define RSSM loss in eq. (9) as a **weighted version of DSM loss**,
> > >
> > > $$
> > > L_{\rm RSSM}(\theta; t, s) = \iint  \underbrace{\exp(Q(s, a_0) / \lambda) \tilde p_t(a_t|s)  \tilde q_{0|t}(a_0 | a_t)}\_{h(a_0, a_t|s)} ||s_\theta(a_t; t, s)-\nabla\_{a_t}\log \tilde q\_{0|t}(a_0|a_t)||^2 da_0da_t
> > > $$
> > >
> > > Then according to Appendix Eq. (31-34, reversely),
> > > $$
> > > L_{\rm RSSM}(\theta; t, s) = \int ||s_\theta(a_t; t, s)-\nabla_{a_t}\log p_t(a_t|s)||^2 g(a_t;s)da_t + \texttt{constant}
> > > $$
> > > where $g$ is strictly positive for all $a_t$. The optimal solution is the same as the  DSM loss,
> > > $$s_\theta(a_t; t, s)=\nabla_{a_t}\log p_t(a_t|s),\forall a_t.$$
> > > Therefore, RSSM loss is equivalent to DSM loss in training diffusion models $s_\theta$.
> > >
> > > **Regarding the confusion about semantic meanings and Bayes' rule.** We changed the weight function from $p(a_0,a_t|s)$ to $h(a_0,a_t|s)$. The reverse sampling trick intends to fit **a different joint distribution embedded in $h$**, so that we can bypass the intractable posterior $q_{0|t}$ associated with the original joint distribution $p$. The change of RSSM is on joint distributions, which do not carry semantic implications.
> > >
> > > **Regarding the `replace` statement**: The `replace` only occurs in the derivations of the optimal solution (Eq. 31-34) , not in the design of the sampling rule of RSSM loss $(**)$. The RSSM loss function is based on the joint distribution  $\tilde p_t(a_t|s)  \tilde q_{0|t}(a_0 | a_t)$, which reflects our intended reverse sampling design.
> > >
> > > We will change our derivations following this reply in the revision to avoid confusion.
> > >
> > > ## 2. Performance claim
> > >
> > > Our original claim in the manuscript is
> > >
> > > > … improves more than 120% over soft actor-critic on complex locomotion tasks such as Humanoid and Ant.
> > >
> > > which is only on complex tasks. Humanoid and Ant are tasks with the highest dimensions (376 and 111 in observations) in Gym MuJoCo.
> > >
> > > Nevertheless, we will change our statement to `improves more than 100% over soft actor-critic on Gym MuJoCo Ant` in the revision to avoid possible overclaim and support the new setup with 1M updates.
> > >
> > >
> > > ## 3. The DIPO statement
> > >
> > > We would like to kindly remind the reviewer that,
> > >
> > > > *Update actions stored in the replay buffer*
> > >
> > > is not practical. The reason is that if we update $a\to a^*$ in the replay buffer $(s, a, r, s')$, then the $r$ is not the reward induced by $s, a^*$ and $s'$ is not a sample from $P(\cdot |s, a^*)$.  The updated tuple $(s, a^*, r, s')$ cannot be used for policy evaluation anymore.
> > >
> > > Therefore, we need to store the updated actions $(s, a^*)$ somewhere else. We will change to `DIPO updates actions stored in a separate diffusion buffer` to be more accurate.
> > >
> > > ----
> > >
> > > We hope the response can further address your concerns. Please feel free to update your response if you have further questions, and we will also update this reply accordingly.

---

### Official Review · Reviewer_pogC · 2025-03-20

**Overall Recommendation:** 3

**Summary:**

This paper highlights the challenges in mimicking an energy-based policy, primarily due to two key reasons: the intractability of the energy function caused by the partition term and the inherent limitation of online RL, where optimal policies are not directly accessible. While existing online diffusion policy algorithms attempt to address these issues, some other limitations remain. In this line, the authors propose RSSM, which samples a one-step estimate of the denoised action to precisely follow the score of the energy-based policy, along with SDAC, a novel diffusion policy algorithm built upon RSSM. The effectiveness of SDAC is demonstrated through experiments conducted on both toy examples and Gym benchmarks.

**Claims And Evidence:**

While most claims appear to be well-founded, the derivation of RSSM is somewhat unclear, which could impact the overall clarity and validity of the statement. Providing a more detailed and structured explanation of this derivation would help strengthen the argument.

**Essential References Not Discussed:**

One of the key contributions is exponential Q-weighted score matching, which closely resembles QIPO (https://openreview.net/pdf?id=HA0oLUvuGI), published in ICLR 2025. While there are differences in application areas (online RL vs. offline RL) and specific methodologies (reverse sampling vs. forward sampling), the fundamental idea appears to be quite similar.

**Experimental Designs Or Analyses:**

I reviewed most aspects of the experimental design and analysis, including the experimental setups and ablations. Most parts of the main experiments seem appropriate, but the use of five parallel online environments for sampling is relatively uncommon. Nonetheless, since the authors have provided baseline performance under the same settings, fairness does not appear to be a significant concern. The toy experiments and ablation studies are generally well-designed, though an additional ablation study for the entropy coefficient would be beneficial. Moreover, given that QSM inevitably differs in its derivation from the original paper, both in implementation details (ex) transitioning from DDPM to Langevin) and hyperparameters, it would be better to carefully verify these aspects to ensure consistency.

**Methods And Evaluation Criteria:**

The ten Gym environments used for evaluation are appropriate and commonly used.

**Other Comments Or Suggestions:**

N/A

**Other Strengths And Weaknesses:**

Based on the provided code, there are some implementation components that do not fully align with or are not explicitly disclosed in the explanations presented in the paper. These include:

1. The implementation involves sampling multiple actions and selecting the one with the highest Q value for both online interaction and evaluation. This approach is referred to as "efficient behavior policy" in the QVPO paper and is also widely adopted in diffusion policy algorithms for offline RL, as initially introduced by the IDQL paper. Disclosing this aspect is important, not only to ensure fairness in evaluation but also because the impact of exploratory sampling, i.e. adding Gaussian noise appears to be significantly influenced by this choice.
2. Critic update. The paper states that SDAC employs soft policy evaluation. However, this does not seem feasible for diffusion policies due to the intractability of the log probability. Additionally, the provided code uses a standard TD update without incorporating log probability terms. Clarifying this discrepancy would help us understand the method more accurately.
3. Entropy update. The provided code indicates that the entropy coefficient $\lambda$ is updated via gradient descent using the difference between the target entropy and the entropy of $\mathcal{N}(\mu, (0.1\lambda)^2 I)$, which corresponds to the distribution of online samples induced by exploratory sampling for given action from the diffusion policy. Notably, $ \lambda $ at each timestep follows a scheduled approach rather than an adaptive one. Additionally, in line 13 of Algorithm 1, even when using the simplified notation, it seems more appropriate to use $\lambda_{e-1}$ instead of $\lambda_e$ immediately after $\beta$. Clarifying these aspects would improve the transparency and accuracy of the explanation.

**Questions For Authors:**

1. Can you clarify or provide additional details on the points mentioned in "Other Strengths and Weaknesses" that you consider essential?
2. Can you provide a more comprehensive and detailed proof for the transition from eq. (13) to eq. (15) to improve clarity and understanding?
3. Since the benchmark settings, using five parallel online environments for sampling over 200K steps, is relatively uncommon, can you provide partial results using a single online environment with 1M steps for comparison?

**Relation To Broader Scientific Literature:**

The key contribution of SDAC, particularly through its derivation from RSSM, is its ability to leverage the gradient of the Q function using approximately denoised actions when the score function is well-aligned. In contrast, most diffusion policy algorithms designed to follow energy-based policies inherently rely on the gradient of the Q function computed with noisy actions. However, in most cases, only the standard Q function is accessible, which may not effectively handle such noisy actions.

**Theoretical Claims:**

I checked all of them and they seem mostly sound. However, I remain uncertain whether the reverse sampling trick can be applied in this manner, specifically from eq. (13) to eq. (15). It would be helpful if the proof for this part were presented in a more comprehensive and detailed manner.

---

> ### Author Rebuttal · Authors · 2025-04-01
>
> We thank the reviewer for the thoughtful and constructive feedback. Below, we first provide detailed derivations of the reverse sampling trick, followed by detailed answers to the specific questions.
>
> ## Reverse Sampling Trick Derivations
>
> Check [rebuttal_proof](https://anonymous.4open.science/r/rebuttal_pdf-D49E/) if the equations do not show up.
>
> We first explain (14) from Gaussian probability density function (PDF),
>
> $\tilde{q}\_{0|t}(\\tilde a\_0|a\_t)$ **is defined as** $\mathcal{N}(\tilde a_0;\frac{1}{\sqrt{\bar\alpha_t}}a_t, \frac{1-\bar\alpha_t}{\bar\alpha_t}I)$ with PDF
>
> $$
> \tilde{q}_{0|t}(\tilde a_0|a_t)  =  (2\pi\frac{1-\bar\alpha_t}{\bar\alpha_t})^{-d/2}\exp(-\frac{||\tilde a_0 - \frac{1}{\sqrt{\bar\alpha_t}}a_t ||^2}{2\frac{(1-\bar\alpha_t)}{\bar\alpha_t}})
> $$
>
> The PDF of forward process perturbation kernel $q_{t|0}(a_t|a_0)$ is
>
> $$
> q_{t|0}(a_t|a_0)=\mathcal{N}(a_t;\sqrt{\bar\alpha_t}{a_0}, (1-\bar\alpha_t)I)  = (2\pi(1-\bar\alpha_t))^{-d/2} \exp(-\frac{||\sqrt{\bar\alpha}_ta_0 - a_t ||^2}{2(1-\bar\alpha_t)})
> $$
>
> **Note that $\tilde q_{0|t}$ is NOT the posterior distribution of $q_{t|0}$, which is usually intractable.**
> Notice that
>
> $$
> \\nabla_{a_t}\log \\tilde{q}\_{0|t}(a_0|a_t) = \\nabla_{a_t}\\log q_{t|0}(a_t|a_0) = - \\frac{a_t - \\sqrt{\bar\\alpha_t}a_0}{1 - \\bar\\alpha_t} \\quad (*)
> $$
>
> With these properties, we start from (13)
>
> $$
> \iint \tilde{p}\_t(a_t | s) q\_{t | 0}(a_t | a_0) \exp (Q(s, a_0) / \lambda) ||s_\theta(a_t;s,t) - \nabla_{a_t}\log q_{t|0}(a_t|a_0)||^2 d a_0 d a_t\quad (13)
> $$
>
> We first substitute in the PDF of $q_{t|0}$,
>
> $$
> = \iint \tilde{p}\_t(a_t \mid s) (2\pi(1-\bar\alpha_t))^{-d/2} \exp(-\frac{||a_t - \sqrt{\bar\alpha_t}a_0||^2}{2(1-\bar\alpha_t)})\exp (Q(s, a_0) / \lambda) ||s_\theta(a_t;s,t) - \nabla_{a_t}\log q_{t|0}(a_t|a_0)||^2 d a_0 d a_t
> $$
>
> $$
> = (\bar\alpha_t)^{d/2}\iint \tilde{p}\_t(a_t \mid s) \underbrace{ (2\pi\frac{1-\bar\alpha_t}{\bar\alpha_t})^{-d/2}\exp(-\frac{||a_0 - \frac{1}{\sqrt{\bar\alpha_t}}a_t ||^2}{2\frac{(1-\bar\alpha_t)}{\bar\alpha_t}}) }_{ \tilde{q}\_{0|t}(a_0|a_t)}\exp (Q(s, a_0) / \lambda) ||s\_\theta(a_t;s,t) - \nabla\_{a_t}\log q\_{t|0}(a_t|a_0)||^2 d a_0 d a_t
> $$
>
> $$
> = (\bar\alpha_t)^{d/2}\iint \tilde{p}\_t(a_t | s) \tilde{q}\_{0|t}(a_0|a_t) \exp (Q(s, a_0) / \lambda)  ||s_\theta(a_t;s,t) - \nabla_{a_t}\log \tilde{q}_{0|t}(a_0|a_t)||^2 d a_0 d a_t \quad (**)
> $$
>
> where the final equality we leverage $(*)$.
>
> The $(**)$ equals the equation (15) times constant $ \bar\alpha_t^{-d/2}$, not affecting the optimal solution at given $t$.
>
> [1] revealed that ignoring the $t$-scale constant weights does not affect the performance of diffusion models when learned jointly like (16), thus we ignore the constant and sample uniformly from timestep $t$.
>
> [1] Ho, J., Jain, A., & Abbeel, P. Denoising diffusion probabilistic models.
>
> ---
>
> ### Experimental results or analysis
>
> Please refer to [Figure 6](https://anonymous.4open.science/r/rebuttal_pdf-D49E/rebuttal_figures.pdf) for ablation study of $\lambda$ learning rate.
>
> ---
>
> ### Essential References Not Discussed
>
> **Q2:** Relation to QIPO.
>
> **A2:** Our loss function looks similar to QIPO. However, the partition function estimation in QIPO is expensive and biased. The proposed SDAC does not need partition function estimation, and our estimation is unbiased.
>
> Moreover, the QIPO paper was published **after** ICML submission, we will add QIPO to the revision.
>
> ---
>
> ### Other Strengths And Weaknesses
>
> **Q3:** "efficient behavior policy" trick:
>
> **A3:** Yes, we use it. We have the same observation as QVPO, the diffusion policy is too random, thus we use the efficient behavior policy trick, sampling multiple actions and selecting the one with the highest Q-function. It is a standard trick in diffusion policy and diffusion-based planners. We skip it due to space limit and will clarify it in the revision.
>
> ---
>
> **Q4:**
> > Log probability is intractable in diffusion policy, ..., standard PD update without incorporating log probability terms.
>
> **A4:** As Q3 mentioned, we use the efficient behavior policy trick and additive Gaussian noise. We notice the policy stochasticity is low after the efficient behavior policy trick, thus we can approximate the entropy using only the log density of the additive Gaussian. We provide results fixing the log density term in [Table 3](https://anonymous.4open.science/r/rebuttal_pdf-D49E/rebuttal_figures.pdf).
>
> ---
>
> **Q5:** Entropy update.
>
> **A5:** Our $\lambda$ updates are adaptive in line 185 of `/relax/algorithm/sdac.py`. We will fix the typo.
>
> **Q6:** Performance with 1M updates
>
> **A6:** We provide 4 envs that are not saturated with 200K updates in [Figure 1](https://anonymous.4open.science/r/rebuttal_pdf-D49E/rebuttal_figures.pdf). SDAC outperforms all the baselines.
>
> ----
>
> We appreciate your thoughtful review and hope our responses have clarified the concerns raised. We would be grateful if you would consider updating your score in light of these clarifications.

---

> > ### Comment · Reviewer_pogC · 2025-04-04
> >
> > Thank you for your response. I have updated my recommendation accordingly.

---

> > > ### Author Response · Authors · 2025-04-08
> > >
> > > Thanks for your reply and updating the score! We are glad that our response clarifies your concerns.
> > >
> > > We will add the explanations to the revision to make it more clear.

---

### Official Review · Reviewer_VLk9 · 2025-03-21

**Overall Recommendation:** 3

**Summary:**

The paper proposes a novel method that leverages diffusion models to enhance SAC, but in a nontrivial way. To address the challenges of using diffusion policies—such as the need to track gradients through the entire reverse chain—the paper introduces RSSM, a new approach for estimating the score function.

The key idea is to reverse the standard diffusion sampling process. Instead of training a conventional diffusion model that samples from $p(a_0)$ and then generates $p(a_t | a_0)$, the proposed method samples from $p(a_t)$ and then estimates $p(a_0 | a_t)$. The paper also provides a theoretical justification that, under the RSSM formulation, it is still valid to learn the score function.

**Claims And Evidence:**

All claims are well supported by rigorous proofs and illustrative examples.

**Essential References Not Discussed:**

None.

**Experimental Designs Or Analyses:**

I have a few minor questions regarding the algorithm and experimental setup:

1. Since action generation still requires reverse sampling, is it possible to directly use $\tilde{q}(a_0 | a_t)$ instead?

2. When normalizing Q-values, do you sample multiple actions for a given state, or is the normalization performed within a batch?

3. I noticed that DACER exhibits extremely high variance in your experiments. However, this phenomenon was not present in the original DACER paper, which used the v3 dataset. Do you have any insights into why this discrepancy occurs? Could you also provide some experimental results on the v3 dataset for a more straightforward comparison?

**Methods And Evaluation Criteria:**

The benchmarks used in the paper are standard and well-accepted.

**Other Comments Or Suggestions:**

None.

**Other Strengths And Weaknesses:**

The paper is easy to follow, and the proposed approach is novel. For potential weaknesses and questions, please refer to my previous section.

**Questions For Authors:**

None.

**Relation To Broader Scientific Literature:**

None.

**Theoretical Claims:**

I reviewed the theoretical proofs and did not notice any apparent issues. The theoretical results also make sense to me. However, I do have a few minor questions:

1. How should $\tilde{p}_t$ be chosen? Specifically, what is the appropriate distribution over $t$, and what form should $\tilde{p}_t$ take? From the current description, it seems that any choice might work for the algorithm—is that correct?

2. Is the assumption in Eq. (10) too strong? Does it limit the model’s expressiveness? Furthermore, does this assumption still yield a valid ELBO? In classical diffusion models, the assumption is typically on $q(a_{t-1} | a_t)$ being Gaussian, not $q(a_0 | a_t)$. I’m concerned that this assumption may be too restrictive.

3. How is entropy estimated using this method? After learning the score function with RSSM, do we still need to integrate along the ODE to compute entropy? Since entropy is required to learn the Q-function, clarification on this step would be helpful.

---

> ### Author Rebuttal · Authors · 2025-04-01
>
> We thank the reviewer for the thoughtful comments. Here are the responses to the questions,
>
> ## Theoretical claims
>
> **Q1:**
>
> > *How should $\tilde{p_t}$ be chosen? Specifically, what is the appropriate distribution over $t$, and what form should $\tilde{p_t}$ take? it seems that any choice might work for the algorithm—is that correct?*
>
>  **A1:** Yes. Theoretically, as long as the support of $\tilde{p}_t$ covers the support of $p_t(a_t)$, i.e., the distribution of the forward process at time $t$, we can recover the noisy score function by optimizing the RSSM loss (9).
>
>  We discussed our empirical choice of $\tilde{p}_t$ in Section 4.1. In online RL, RSSM loss depends on the Q-function, which might not be accurate for any action $a$. Therefore, we design $\tilde{p}_t$ by sampling from the reverse process induced by the current policy, ensuring alignment with regions where the Q-function is more reliable.
>
>   ----
>
> **Q2:**
> > *Is the assumption in Eq. (10) too strong? Does it limit the model’s expressiveness? Furthermore, does this assumption still yield a valid ELBO? ...*
>
> **A2:**
> We would like to clarify that the reverse sampling distribution $\tilde q_{0|t}(a_0|a_t)$ in eq.(10) is not the unknown posterior distribution $q_{0|t}(a_0|a_t)$ in the reverse process, or its approximations.
> We did not change the reverse process, so its expressiveness and the ELBO to learn it are not affected.  $\tilde{q}_{0|t}(a_0|a_t)$ is a known distribution $\mathcal{N}(\frac{1}{\sqrt{\bar{\alpha_t}}}a_t, \frac{1-\bar\alpha_t}{\alpha_t}I)$, only used in computing RSSM loss function.
>
> The proposed RSSM loss and original DDPM loss to train diffusion models [1] are both derived from the same ELBO. However, DDPM loss is not tractable when we cannot sample from dataset $p_0$ and only know its energy function, which is the case in online RL. The novelty of RSSM is a tractable sampling-based method to train a diffusion model/policy when only the energy function is available.
>
>  ----
>
> **Q3:**
> > *How is entropy estimated using this method?*
>
> **A3**:  When we implement the diffusion policy, we leverage a trick called efficient behavior policy, where we sample multiple actions and choose the one with the highest $Q$ function, which is also used in many diffusion policy papers like [1,2]. We notice that after the efficient behavior policy trick, the policy stochasticity is low, so we can estimate the entropy by only the entropy of additive Gaussian noise.
>
> Our strong empirical performance shows that this simple entropy estimation works well enough.
>
> [1]
> Ding, Shutong, et al. Diffusion-based reinforcement learning via q-weighted variational policy optimization.
>
> [2] Janner, Michael, et al. Planning with diffusion for flexible behavior synthesis.
>
>  ----
>
> ## Experimental Designs
>
> **Q4:**
> > *Clarification of reverse sampling distribution $\tilde{q}(a_0 | a_t)$ since action generation still requires reverse sampling, is it possible to directly use $\tilde{q}(a_0 | a_t)$ instead?*
>
> **A4:** As the **A2** pointed out, $\tilde{q}_{0|t}$ is not the reverse process posterior, and we did not change the reverse denoising process. Therefore, we still need to sample from the reverse process, which is the key to the rich expressiveness of diffusion models.
>
> ----
>
> **Q5:**
> > *When normalizing Q-values, do you sample multiple actions for a given state, or is the normalization performed within a batch?*
>
> **A5:** We just normalize within the batch, no additional sampling is needed.
>
> ----
>
> **Q6:**
> > *I noticed that DACER exhibits extremely high variance in your experiments. However, this phenomenon was not present in the original DACER paper, which used the v3 dataset. Do you have any insights into why this discrepancy occurs? Could you also provide some experimental results on the v3 dataset for a more straightforward comparison?*
>
> **A6:** Thank you for the keen observations. The differences stem from hyperparameter choices: the original DACER uses a smaller learning rate (1e-4) and 20 parallel environments, resulting in significantly more fresh data and lower variance. In contrast, we use 3e-4 with 5 environments across all algorithms for a fair comparison.
>
> We provide DACER (using both our and original hyperparameters) and SDAC in both V3 and V4 in [Figure 2 and Table 2](https://anonymous.4open.science/r/rebuttal_pdf-D49E/rebuttal_figures.pdf). The original DACER parameters show great stability but much slower learning. The variances of V3 and V4 envs are similar.
>
> The proposed SDAC can achieve stable learning with a larger learning rate and fewer envs (as 1 is commonly used), demonstrating better robustness compared to DACER.
>
> ----
>
> We appreciate your thoughtful review and hope our responses have clarified the concerns raised. We would be grateful if you would consider updating your score in light of these clarifications.

---

> > ### Comment · Reviewer_VLk9 · 2025-04-05
> >
> > Thank you for the authors' detailed response. My concerns regarding expressiveness and the ELBO have been addressed.
> >
> > However, I still have questions about the role of $\tilde{p}(t)$. You mention that it can be chosen arbitrarily, but shouldn't it satisfy the marginal distribution $p_t = \int \pi(a_0) p(a_t \mid a_0)$? Only in this case will the joint distribution of the forward process, $p(a_0, a_t) = \pi(a_0)p(a_t \mid a_0)$, match your RSSM sampling procedure, where $p(a_0, a_t) = \pi(a_t)p(a_0 \mid a_t)$, and Eq. (17) is used to sample from $p(a_0 \mid a_t)$ — which seems valid to me now.
> >
> > That said, I wonder if sampling from $\pi(a_t)$ is as difficult as sampling from $\pi(a_0)$? If so, doesn't this reintroduce the original challenge?

---

> > > ### Author Response · Authors · 2025-04-06
> > >
> > > Thank you for the comments. We are glad that our reply has addressed your concerns.
> > >
> > > Regarding the arbitrary choice of $\tilde p_t$, the short answer is:
> > >
> > > The RSSM loss function in (9) is not an expectation over the joint distribution $p(a_0, a_t)$. By definition, **RSSM multiplies this joint distribution $p(a_0, a_t)$ by weights $\tilde p_t(a_t|s)Z(s)$ with $Z(s) = \int \exp (Q(s, a_0)/\lambda)da_0$, while maintaining the same optimal solution $s_\theta(a_t;t,s) = \nabla_{a_t}\log\pi_t(a_t|s)$.**
> > >
> > > After weighting, the new measure becomes
> > > $$
> > > \tilde p_t(a_t|s)Z(s) p(a_0,a_t | s) \propto \exp(Q(s, a_0)/\lambda)\tilde p_t(a_t|s) \tilde q_{t|0}(a_t|a_0),
> > > $$
> > > which admits our reverse sampling in equation (16)-(17) with arbtrary $\tilde p_t$ as unbiased sampling-based approximations.
> > >
> > > This weighting technique is the core design of RSSM, and we explain the details in the following.
> > >
> > > ---
> > >
> > >
> > > ## Core of our RSSM loss design: Reweighting on $a_t$ without changing the optimal solutions
> > >
> > > ### Original DSM loss and optimal solution
> > >
> > > We start from the denoising score matching (DSM) loss in [1] (also equation (5)) at given $t$ and $s$,
> > > $$
> > > L_{\rm DSM}(\theta; t, s) = \iint ||s_\theta(a_t; t, s)-\nabla_{a_t}\log q_{t|0}(a_t|a_0)||^2 p(a_0,a_t | s)da_0da_t \quad (*)
> > > $$
> > > which exactly needs to sample from the joint distribution $p(a_0,a_t | s)$ you mentioned.
> > >
> > > According to our derivation in the Appendix, Proposition 1 (line 606-617), we can show that the DSM loss can be written in an equivalent form:
> > > $$
> > > L_{\rm DSM}(\theta; t, s) = \int ||s_\theta(a_t; t, s)-\nabla_{a_t}\log p_t(a_t|s)||^2 p_t(a_t|s) da_t +\texttt{constant}
> > > $$
> > > where the optimal solution is achieved when $s_\theta$ matches the empirical score function for all $a_t$:
> > > $$
> > > s_\theta(a_t; t, s)=\nabla_{a_t}\log p_t(a_t|s),\forall a_t
> > > $$
> > >
> > > ### RSSM as weighted DSM loss
> > >
> > > Note that we can compute a weighted version of Equation $(\*)$ by multiplying aribtary strictly positive weight functions of $a_t$ with $p_t(a_t|s)||s\_\theta(a\_t; t, s)-\nabla\_{a\_t}\log p\_t(a\_t|s)||^2$ to define a different loss function. This weighted DSM loss admits the same optimal solution as the (unweighted) DSM loss in Equation $(*)$.
> > >
> > > The proposed RSSM loss chooses the weight function to be $\tilde{p}_t(a_t|s) Z(s)p_t(a_t|s)$, as we derive in Equation (12) and (13),
> > >
> > > $$
> > > L_{\rm RSSM}(\theta; t, s) = \int ||s_\theta(a_t; t, s)-\nabla_{a_t}\log p_t(a_t|s)||^2  p(a_t | s) \underbrace{\tilde  p_t(a_t|s)Z(s)}_{\text{RSSM weights}}da_t
> > > $$
> > >
> > > where $\tilde p_t$ is the sampling distributoin we can choose and $p_t(a_t|s) = \int \pi(a_0|s)q_{t|0}(a_t|a_0)da_0$ is the $t$-step noise-perturbed policy. $\tilde p_t$ and $p_t$ can be different.
> > >
> > > We can see that its optimal solution is still $s_\theta(a_t; t, s)=\nabla_{a_t}\log p_t(a_t|s),\forall a_t$ if $\tilde p_t$ has full support.
> > >
> > > Then, according to Appendix Equation (31)-(34),
> > > $$
> > > L_{\rm RSSM}(\theta; t, s) = \iint  \exp(Q(s, a_0) / \lambda) \tilde p_t(a_t|s)  \tilde q_{0|t}(a_0 | a_t) ||s_\theta(a_t; t, s)-\nabla_{a_t}\log \tilde q _{0|t}(a_0|a_t)||^2 da_t + \texttt{constant}
> > > $$
> > >
> > > which is equivalent to the RSSM loss function in (9), and we can use reverse sampling with arbitrary $\tilde p_t$, rather than the $p_t$ that is not available to sample from.
> > >
> > > ---
> > > We hope this further explanation clarifies your concern. Please feel free to update your comments if you have further questions, we will update our reply accordingly.

---

### Decision · Program_Chairs · 2025-05-01

**Decision:**

Accept (poster)

**Comment:**

This paper presents SDAC, a novel diffusion-policy method for online reinforcement learning. The approach avoids unstable Q score matching and costly multi-step backpropagation by introducing reverse sampling, a technique that nonparametrically maps noisy actions to the denoised domain. The paper establishes a theoretical connection between reverse sampling score matching (RSSM) and regression to an energy-based policy under a custom measure, supported by a toy example using a Gaussian mixture target. SDAC also demonstrates strong empirical performance on the MuJoCo benchmark.

The paper's key strengths include consistent empirical improvements across diverse tasks, a simple yet novel core methodology, and solid theoretical grounding. However, some limitations remain, such as limited clarity in parts of the theoretical derivation, missing implementation details, and minor inaccuracies in the discussion of related work.

The rebuttal addresses most concerns through detailed explanations and additional experiments. These clarifications should be sufficiently reflected in the final version. In particular, the authors are encouraged to clearly articulate the role of reverse sampling and the custom measure, include missing implementation details, such as QVPO-style efficient behavior, present key results in the main text, and carefully verify baseline information by consulting both papers and official codebases, such as DQS and QSM.

Overall, this paper is recommended for acceptance, provided that the raised issues and limitations are incorporated into the final version.